# ∞ InfVSR:
# Toward Consistency-Driven Streaming Generative Video Super-Resolution

**Ziqing Zhang**[* 1] **Kai Liu**[* 1] **Zheng Chen**[1]
**Xi Li**[2] **Yucong Chen**[2] **Bingnan Duan**[2] **Linghe Kong**[† 1] **Yulun Zhang**[† 1]

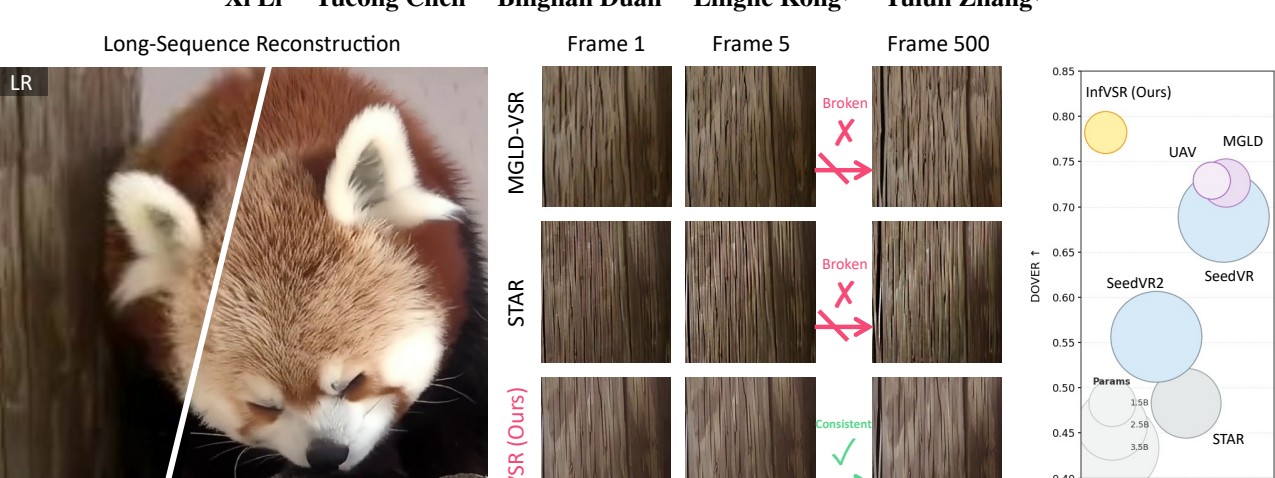

*Figure 1.* Speed and multi-frame comparisons. Our InfVSR is capable of seamlessly and streamingly upscaling videos with unbounded length, and demonstrates the best quality and fastest speed among existing diffusion-based methods.

## Abstract

Real-world videos often extend over thousands of frames. Existing generative video super-resolution (VSR) approaches, however, face two persistent challenges when processing long sequences: (1) *inefficiency* due to the heavy cost of multi-step denoising for full-length sequences; and (2) poor *consistency* hindered by temporal decomposition that causes artifacts and discontinuities. To break these limits, we propose **InfVSR**, which reformulates VSR as an **autoregressive-one-step-diffusion** paradigm, and enables streaming inference with video diffusion priors. First, we adapt the pretrained DiT into a causal structure, maintaining both local and global coherence via rolling KV-cache and joint visual guidance. Second, we distill the diffusion process into a single step efficiently, with patch-wise pixel supervision and cross-chunk distribution matching.

To fill the gap in long-form video evaluation, we build a new benchmark tailored for extended sequences and further introduce semantic-level metrics to comprehensively assess temporal consistency. Our method pushes the frontier of long-form VSR, achieves state-of-the-art quality with enhanced semantic consistency, and delivers up to **58×** speed-up over existing methods such as MGLD-VSR. Our code and models are available at https://github.com/Kai-Liu001/InfVSR.

## 1. Introduction

The real world is witnessing an explosive growth in video content, driving an increasing demand for high-quality visual experiences. Video super-resolution (VSR), which aims to reconstruct high-resolution (HR) videos from low-resolution (LR) inputs, has therefore become a fundamental task for content restoration (Chen et al., 2026; Wang et al., 2026a), streaming (Shiu et al., 2025; Zhuang et al., 2025), and archival applications (Liu et al., 2026; Li et al., 2026a). In recent years, generative models, particularly diffusion (Ho et al., 2020; Song et al., 2021)-based ones, are most popular and preferred. With powerful pretrained priors, they can recover realistic details under complex real-world degradations (Wang et al., 2021; Chan et al., 2022b).

Despite recent advances (Xie et al., 2025; Wang et al., 2025a;

[1]Shanghai Jiao Tong University [2]Meituan Inc. Correspondence to: Yulun Zhang[†] <yulun100@gmail.com>, Linghe Kong[†] <linghe.kong@sjtu.edu.cn>.

*Proceedings of the 43rd International Conference on Machine Learning*, Seoul, South Korea. PMLR 306, 2026. Copyright 2026 by the author(s).

Chen et al., 2025b; Wang et al., 2025b; 2026b), translating such generative capability into practical, large-scale video enhancement systems remains challenging. A central bottleneck lies in **maintaining temporal consistency under realistic efficiency constraints**. Unlike image super-resolution (ISR), VSR must ensure temporal consistency across frames. Inconsistent details such as flickering textures or unstable colors, while not reflected by perceptual metrics like (Ke et al., 2021; Wang et al., 2023a; Liu et al., 2024b), can severely disrupt the viewer's experience. This issue is amplified when processing long videos under strict memory and latency constraints (Liu et al., 2024a; 2025b;a), because temporal decomposition into short chunks is essential, but breaks global temporal coherence learned by pretrained models (Du et al., 2025; Li et al., 2026b). Existing solutions like increased overlap or post-hoc smoothing (Wang et al., 2024; 2025b) only partially address this, leaving long-range and semantic-level consistency unpreserved. Thus, there is a pressing need for VSR models that are (1) **streamable**, efficiently handling arbitrary-length videos with fixed memory, and (2) **consistent**, explicitly modeling and enforcing temporal consistency apart from fidelity and visual quality.

Among the first explorations, we propose **InfVSR**, the first consistency-driven streaming generative VSR framework. While powerful pretrained video generation priors provide our foundation, our work focuses on and explores consistency, while maintaining strong performance and efficiency, through the following three key aspects:

**(1) Modeling consistency with AR-OSD reformulation.** Our starting point is the recognition that, compared to post hoc learned alignments, video diffusion priors inherently capture robust and generalizable temporal distribution. Then to enable streaming, we reformulate video super-resolution as an autoregressive one-step diffusion (AR-OSD) process, where **intra-chunk diffusion** refines each chunk in one forward pass, and **inter-chunk autoregression** propagates temporal consistency across chunks. To preserve both short- and long-range consistency, we implement a dual-timescale conditioning mechanism. A **rolling KV-cache** embedded in the self-attention layers captures local transitions across chunk boundaries, ensuring smooth motion and appearance changes. Meanwhile, a **joint visual guidance** module extracts global semantic anchors from reference LR frames and injects them into the cross-attention layers, stabilizing long-range semantics. This design allows the model to maintain strong temporal coherence while operating with constant memory and without revisiting previous frames.

**(2) Enforcing consistency with dual-level constraints.** We adopt a consistency-aware training scheme that combines local fidelity with global temporal regularization. First, a **patch-wise pixel supervision** strategy enables efficient high-resolution learning via spatially sampled HR ground truth patches, which drastically reduces memory usage com-

pared to full-frame supervision. This also ensures frame-level temporal constraints in the pixel space. Second, a **cross-chunk distribution matching** (Yin et al., 2024b;a) loss aligns the feature statistics between teacher and student generations across autoregressive steps. This constraint provides alignment at the video distribution level, leading to empirically higher semantic consistency. To improve training stability, we apply a two-stage curriculum: the model is first trained to match the HR target in a single step, and then gradually introduced to full autoregressive inference with both cache and prompts enabled.

**(3) Benchmarking consistency with richer dataset and metrics.** To systematically evaluate temporal coherence under long-form settings, we establish **MovieLQ**, a benchmark of ten 1000-frame-long real-world-degraded videos. Beyond conventional pixel-wise metrics ($E^*_{warp}$), we adopt semantic-level temporal metrics from VBench (Huang et al., 2024) to quantify identity consistency, motion smoothness, and background stability over long sequences, providing a robust testbed for streaming-compatible VSR methods.

Together, with these designs, InfVSR achieves SOTA metric, visual quality and long-term consistency, as shown in Fig. 1. Moreover, in efficiency, it delivers up to 58× speed-up over existing methods such as MGLD-VSR (Yang et al., 2024). Overall, our contributions can be summarized as follows:

- We propose **InfVSR**, which is among the first T2V-based autoregressive-one-step-diffusion framework for real-world VSR, supporting ultra-efficient inference on unbounded-length sequences.

- For the AR framework, we introduce a dual-timescale mechanism with rolling KV-cache and joint visual guidance. For training, we design patch-wise pixel supervision and cross-chunk distribution matching, well-balancing fidelity, consistency, and efficiency.

- We establish MovieLQ, a benchmark tailored for long-form VSR under real-world degradations, and first adopt metrics assessing semantic-level consistency.

- Extensive experiments demonstrate the effectiveness and efficiency of our method. We achieve SOTA performance with significantly reduced latency and cost.

## 2. Related Work

**Video Super-Resolution.** Video super-resolution aims to recover high-quality videos from degraded inputs. Early methods rely on RNNs (Chan et al., 2021; 2022a) or window-based transformers (Li et al., 2020; Yi et al., 2019) trained on synthetic degradation, which limits their performance in real-world scenarios. Later works (Chan et al., 2022b; Yang et al., 2021) introduce mixed degradations such as blur and compression, but still struggle to produce sharp and stable results. Diffusion models (Song et al., 2021; Ho

et al., 2020; Rombach et al., 2022) have recently brought a major shift by introducing strong generative priors. Text-to-Image (T2I) based methods (Rombach et al., 2022; Peebles & Xie, 2023; Esser et al., 2024), such as Upscale-A-Video (Zhou et al., 2024) and MGLD-VSR (Yang et al., 2024), inject optical flow (Ranjan & Black, 2017; Teed & Deng, 2020) or temporal modules into pretrained image diffusion backbones. However, their frame alignment remains fragile and error-prone. Text-to-Video (T2V) (Blattmann et al., 2023; Yang et al., 2025; Wan et al., 2025; HaCohen et al., 2024) based methods like STAR (Xie et al., 2025) and SeedVR (Wang et al., 2025a) leverage video-scale priors and achieve significantly better temporal coherence. Models such as DOVE (Chen et al., 2025b) and SeedVR2 (Wang et al., 2026b) further distill multi-step denoising into one-step inference, offering large speedups. Yet these models still suffer from memory cost scaling with video length and temporal inconsistency caused by independent chunk inference. We address both limitations by reformulating VSR as an autoregressive-one-step-diffusion paradigm, while simultaneously leveraging strong video generative priors.

**Autoregressive Video Generation.** Recent progress in large-scale T2V pretraining (Yang et al., 2025; Kong et al., 2024; Wan et al., 2025) has paved the way for autoregressive video generation, which in turn emphasizes real-time operation and stronger controllability. One line (Kondratyuk et al., 2024; Yuan et al., 2025) adopts GPT-style next-token prediction in latent space. Another (Yin et al., 2025; Teng et al., 2025; Chen et al., 2025a; Zhang et al., 2025) explores AR-diffusion by performing intra-frame denoising and inter-frame rollout, which better matches video dynamics and uses T2V priors. Training strategies include diffusion-forcing (Chen et al., 2024), teacher-forcing (Williams & Zipser, 1989), and self-forcing (Huang et al., 2025). They enable generation conditioned on a clean or low-noise context from previous segments. Our work is, to our knowledge, among the first to explore AR-diffusion models for long-form VSR. And beyond simply applying existing generation techniques, we tailor the design to VSR's LR inputs, high-fidelity demands, and high-resolution training. Also notably, we achieve inference with one forward pass, without multiple sampling steps or an additional pass for clean KV cache.

## 3. Methodology

### 3.1. Problem Formulation

In this work, we make the first attempt to reformulate VSR as an **autoregressive-one-step-diffusion** (AR-OSD) paradigm. Our core idea is to maximally retain the capabilities of pretrained T2V priors, while introducing a new autoregressive formulation for temporal modeling. Specifically, as illustrated in Fig. 2, we divide the video into non-overlapping chunks and model: (1) **intra-chunk diffusion**,

where the generative prior is invoked to refine the current chunk, and (2) **inter-chunk autoregression**, where temporal consistency is preserved by propagating information from past chunks. Formally, let the video be divided into $K$ non-overlapping chunks. For each chunk $k$, let $\mathbf{x}_k$ and $\mathbf{y}_k$ denote the LR input and the HR output, respectively. We factorize the joint distribution autoregressively as:

$$p(\mathbf{y}_{1:K} \mid \mathbf{x}_{1:K}) = \prod_{k=1}^{K} p(\mathbf{y}_k \mid \mathbf{x}_k, \mathcal{P}_k), \tag{1}$$

where $\mathcal{P}_k$ represents the autoregressive context collected from previously generated chunks. Each conditional $p(\mathbf{y}_k \mid \mathbf{x}_k, \mathcal{P}_k)$ is approximated by a one-step diffusion:

$$\mathbf{y}_k = G_\theta(\mathbf{x}_k, \mathcal{P}_k). \tag{2}$$

Here, $G_\theta$ is a generator adapted from a pretrained T2V diffusion backbone, conditioned on the current LR chunk and a compact context $\mathcal{P}_k$ derived from past outputs.

### 3.2. Causal DiT Architecture

Our method builds upon a pretrained T2V diffusion model, which typically consists of: (1) a 3D VAE that compresses spatial dimensions by $8\times$ and temporal length from $4n + 1$ to $n + 1$ frames, and (2) a pretrained 3D DiT. As this DiT is originally pretrained for generating short video clips with bidirectional full attention, adaptations need to be made to support causal inference. To be specific, we introduce a dual-timescale temporal modeling based on VSR specialties.

**Local Smoothness via Rolling KV-cache.** To enhance long-form inference in pretrained DiTs, we implement a KV-cache mechanism inspired by LLMs. By assigning absolute positional embeddings to queries and keys based on their original timeline, the model maintains temporal awareness. Concatenating current KV tensors with cached ones ensures smooth transitions without reprocessing earlier frames. We adopt a rolling cache for VSR because, unlike video generation, VSR benefits from strong structural priors in LR inputs, making extensive memory unnecessary. We adopt a fixed-length, rolling KV-cache for two reasons: Efficiency: It keeps memory and computation constant, preventing unbounded growth. Generalization: It ensures a consistent local distribution during training and inference. This strategy achieves strong temporal consistency and constant memory usage, regardless of video length.

**Global Coherence via Joint Visual Guidance.** While the rolling KV-cache preserves local temporal continuity, cache truncation may still lose long-range cues. In standard video generation, this often necessitates external memory banks (Liang et al., 2025). In VSR, however, the LR video itself provides a strong global prior. We leverage this property by introducing joint visual guidance, which provides (1) a shared semantic anchor across multiple neighboring chunks to enhance consistency, while (2) serving as an extremely lightweight alternative to VLM-based prompt

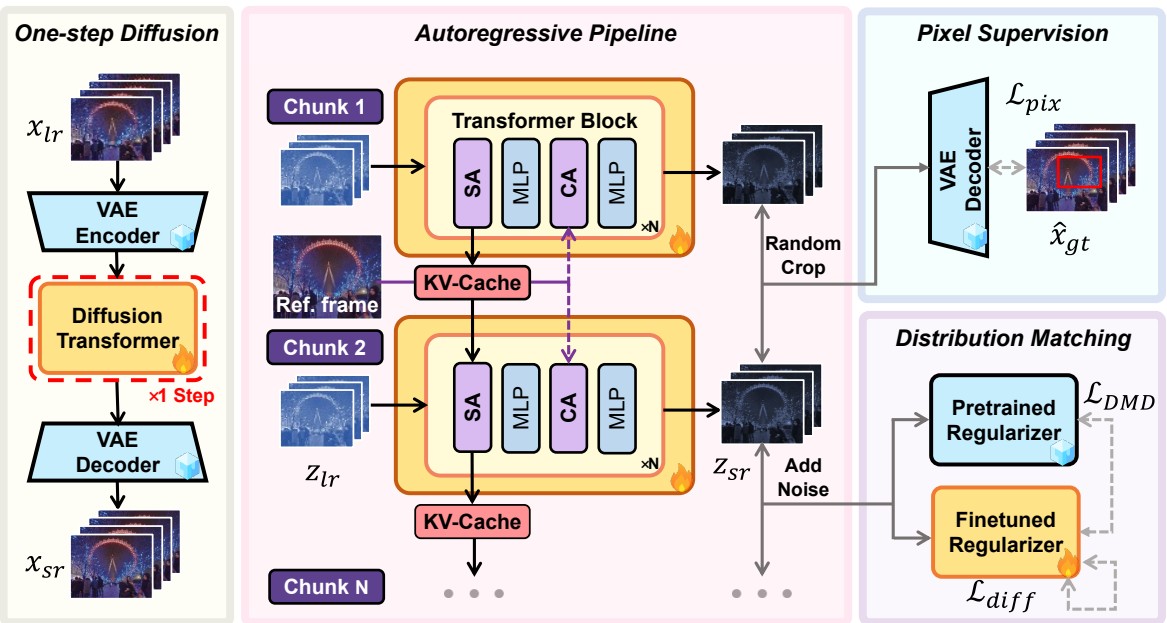

*Figure 2.* Overview of the framework and training strategy of InfVSR. Our method combines intra-chunk **one-step diffusion** with inter-chunk **autoregression** for efficient and scalable VSR. AR is supported by local KV-cache and global joint visual guidance. To enable effective and efficient training, we adopt two objectives: (1) **patch-wise pixel supervision**, which guides detail reconstruction with significantly reduced memory decoding through random spatial cropping; and (2) **cross-chunk distribution matching**, which enforces high-level consistency with a pretrained and a finetuned regularizer, following (Wang et al., 2023c; Yin et al., 2024b; Wu et al., 2024a).

extraction for improving restoration quality. Specifically, LR key frames are encoded with a pretrained visual encoder (i.e., DAPE (Wu et al., 2024b)), and injected as visual prompts into the cross-attention layers of the DiT backbone. These prompts are reused across multiple adjacent chunks to reduce overhead and stabilize context, and are updated through threshold-based key-frame switching when large motion or low reference relevance is detected. By combining rolling KV-cache with joint visual guidance, our framework achieves both local smoothness and global coherence across long sequences, with low computational overhead.

### 3.3. Efficient Autoregressive Post-Training

Training our AR-OSD VSR model requires careful design to balance fidelity, temporal consistency, and memory efficiency. We design a lightweight yet effective training scheme composed of two supervision losses as illustrated in Fig. 2 and a staged curriculum.

**Patch-wise Pixel Supervision.** A pixel-domain reconstruction loss is widely adopted in one-step ISR methods (Wu et al., 2024a; Dong et al., 2025), for recovering high-fidelity visual details. However, in VSR, the $8\times$ spatial and $4\times$ temporal upsampling in the 3D VAE decoder incurs prohibitive memory overhead, which rapidly increases with video resolution. It hinders training DiT at high resolutions, making it difficult to handle long token sequences of VSR tasks.

To address this, we propose **patch-wise pixel supervision**, which is built upon the insight that, in the spatial domain, applying reconstruction loss to randomly cropped video

patches is **expectationally equivalent** to computing the loss over the full video. Specifically, let the predicted latent video $\hat{\mathbf{z}} \in \mathbb{R}^{B \times C \times F \times H \times W}$, and let $D(\cdot)$ be the 3D VAE decoder. We define a random spatial cropping operator $\mathcal{C}_{\text{lat}}(\cdot)$ that extracts a patch of size $h \times w$ from the latent tensor, and another operator $\mathcal{C}_{\text{pix}}(\cdot)$ for the HR video, which extracts the corresponding patch in ground-truth space. Due to the $8\times$ spatial upsampling in $D(\cdot)$, we align the cropping windows such that: $\mathcal{C}_{\text{lat}}$ samples a random crop window at position $(i, j)$ in latent space, and $\mathcal{C}_{\text{pix}}$ applies the same window at position $(8i, 8j)$ in pixel space. Then, the decoded high-resolution patch sequence is:

$$\hat{\mathbf{x}}_{\text{sr}} = D\left(\mathcal{C}_{\text{lat}}(\hat{\mathbf{z}})\right), \quad \hat{\mathbf{x}}_{\text{gt}} = \mathcal{C}_{\text{pix}}(\mathbf{x}_{\text{gt}}), \quad (3)$$

where $\hat{\mathbf{x}}_{\text{sr}}, \hat{\mathbf{x}}_{\text{gt}} \in \mathbb{R}^{B \times 3 \times (4F-3) \times 8h \times 8w}$ are the decoded super-resolved and ground-truth patch sequences, respectively. We then apply two types of loss functions over these cropped patches. The first is a fidelity loss that encourages spatial accuracy and structural realism:

$$\mathcal{L}_{\text{fidel}} = \lambda_{\text{mse}} \cdot \mathcal{L}_{\text{mse}}(\hat{\mathbf{x}}_{\text{sr}}, \hat{\mathbf{x}}_{\text{gt}}) + \lambda_{\text{dists}} \cdot \mathcal{L}_{\text{dists}}(\hat{\mathbf{x}}_{\text{sr}}, \hat{\mathbf{x}}_{\text{gt}}), \quad (4)$$

where $\mathcal{L}_{\text{mse}}$ and $\mathcal{L}_{\text{dists}}$ denote the mean squared error and DISTS (Ding et al., 2020) perceptual loss respectively, and $\lambda_{\text{mse}}$ and $\lambda_{\text{dists}}$ are loss scalers. In addition, we introduce a temporal smoothness loss to enforce local temporal consistency by aligning frame-wise differences:

$$\mathcal{L}_{\text{temp}} = \lambda_{\text{temp}} \cdot \left\| (\hat{\mathbf{x}}_{\text{gt}}^{t+1} - \hat{\mathbf{x}}_{\text{gt}}^{t}) - (\hat{\mathbf{x}}_{\text{sr}}^{t+1} - \hat{\mathbf{x}}_{\text{sr}}^{t}) \right\|_2^2, \quad (5)$$

where $\hat{\mathbf{x}}_{\text{gt}}^{t}$ and $\hat{\mathbf{x}}_{\text{sr}}^{t}$ denote the ground-truth and predicted frames at frame $t$ respectively, and $\lambda_{\text{temp}}$ is the loss scaler. The total patch-wise supervision loss is $\mathcal{L}_{\text{pix}} = \mathcal{L}_{\text{fidel}} + \mathcal{L}_{\text{temp}}$.

*Table 1.* Quantitative comparison with state-of-the-art methods. The best and second performances are marked in red and orange respectively. Our method outperforms on various datasets and metrics.

| Datasets | Metrics | RealBasicVSR CVPR 2022 | RealViFormer ECCV 2024 | Upscale-A-Video CVPR 2024 | MGLD-VSR ECCV 2024 | STAR ICCV 2025 | SeedVR CVPR 2025 | SeedVR2 ICLR 2026 | Ours - |
|---|---|---|---|---|---|---|---|---|---|
| UDM10 | PSNR ↑ | 24.13 | 24.64 | 21.72 | 24.23 | 23.47 | 23.39 | 25.38 | 24.86 |
| | SSIM ↑ | 0.6801 | 0.6947 | 0.5913 | 0.6957 | 0.6804 | 0.6843 | 0.7764 | 0.7274 |
| | LPIPS ↓ | 0.3908 | 0.3681 | 0.4116 | 0.3272 | 0.4242 | 0.3583 | 0.2868 | 0.2972 |
| | DISTS ↓ | 0.2067 | 0.2039 | 0.2230 | 0.1677 | 0.2156 | 0.1339 | 0.1512 | 0.1422 |
| | MUSIQ ↑ | 59.06 | 57.90 | 59.91 | 60.55 | 41.98 | 53.62 | 49.95 | 62.88 |
| | CLIP-IQA ↑ | 0.3494 | 0.4157 | 0.4697 | 0.4557 | 0.2417 | 0.3145 | 0.2987 | 0.5142 |
| | DOVER ↑ | 0.7564 | 0.7303 | 0.7291 | 0.7264 | 0.4830 | 0.6889 | 0.5568 | 0.7826 |
| | $E^*_{warp}$ ↓ | 3.10 | 2.29 | 3.97 | 3.59 | 2.08 | 3.24 | 1.98 | 1.95 |
| SPMCS | PSNR ↑ | 22.17 | 22.72 | 18.81 | 22.39 | 21.24 | 21.22 | 22.57 | 22.25 |
| | SSIM ↑ | 0.5638 | 0.5930 | 0.4113 | 0.5896 | 0.5441 | 0.5672 | 0.6260 | 0.5697 |
| | LPIPS ↓ | 0.3662 | 0.3376 | 0.4468 | 0.3262 | 0.5257 | 0.3488 | 0.3176 | 0.3166 |
| | DISTS ↓ | 0.2164 | 0.2108 | 0.2452 | 0.1960 | 0.2872 | 0.1611 | 0.1757 | 0.1742 |
| | MUSIQ ↑ | 66.87 | 64.47 | 69.55 | 65.56 | 36.66 | 62.59 | 60.17 | 67.75 |
| | CLIP-IQA ↑ | 0.3513 | 0.4110 | 0.5248 | 0.4348 | 0.2646 | 0.3945 | 0.3811 | 0.5319 |
| | DOVER ↑ | 0.6753 | 0.5905 | 0.7171 | 0.6754 | 0.3204 | 0.6576 | 0.6320 | 0.7302 |
| | $E^*_{warp}$ ↓ | 1.88 | 1.46 | 4.22 | 1.68 | 1.01 | 1.72 | 1.23 | 1.25 |
| MVSR4x | PSNR ↑ | 21.80 | 22.44 | 20.42 | 22.77 | 22.42 | 21.54 | 21.88 | 22.49 |
| | SSIM ↑ | 0.7045 | 0.7190 | 0.6117 | 0.7417 | 0.7421 | 0.6869 | 0.7678 | 0.7373 |
| | LPIPS ↓ | 0.4235 | 0.3997 | 0.4717 | 0.3568 | 0.4311 | 0.4944 | 0.3615 | 0.3452 |
| | DISTS ↓ | 0.2498 | 0.2453 | 0.2673 | 0.2245 | 0.2714 | 0.2229 | 0.2141 | 0.2107 |
| | MUSIQ ↑ | 62.96 | 61.99 | 69.80 | 53.46 | 32.24 | 42.56 | 35.29 | 64.03 |
| | CLIP-IQA ↑ | 0.4118 | 0.5206 | 0.6106 | 0.3769 | 0.2674 | 0.2272 | 0.2371 | 0.5229 |
| | DOVER ↑ | 0.6846 | 0.6451 | 0.7221 | 0.6214 | 0.2137 | 0.3548 | 0.3098 | 0.6872 |
| | $E^*_{warp}$ ↓ | 1.69 | 1.25 | 5.07 | 1.55 | 0.61 | 2.73 | 1.08 | 1.03 |
| VideoLQ | MUSIQ ↑ | 55.62 | 52.18 | 55.04 | 51.00 | 39.66 | 54.41 | 39.10 | 56.26 |
| | CLIP-IQA ↑ | 0.3433 | 0.3553 | 0.4132 | 0.3465 | 0.2652 | 0.3710 | 0.2359 | 0.4454 |
| | DOVER ↑ | 0.7388 | 0.6955 | 0.7370 | 0.7421 | 0.7080 | 0.7435 | 0.6799 | 0.7556 |
| | $E^*_{warp}$ ↓ | 5.97 | 4.47 | 13.47 | 6.79 | 5.96 | 9.27 | 8.34 | 7.52 |
| MovieLQ | MUSIQ ↑ | 62.59 | 63.74 | 68.49 | 67.90 | 56.57 | 64.42 | 61.13 | 68.65 |
| | CLIP-IQA ↑ | 0.4672 | 0.4227 | 0.5117 | 0.5591 | 0.3411 | 0.505 | 0.4468 | 0.5888 |
| | DOVER ↑ | 0.8234 | 0.8273 | 0.775 | 0.8402 | 0.7565 | 0.8145 | 0.8031 | 0.8447 |
| | $E^*_{warp}$ ↓ | 3.39 | 2.24 | 5.53 | 3.67 | 3.11 | 4.70 | 4.26 | 2.88 |

**Cross-Chunk Distribution Matching.** While the temporal smoothness loss $\mathcal{L}_{temp}$ encourages frame-to-frame consistency and achieves low $E^*_{warp}$ scores, we ask whether the powerful spatiotemporal priors of pretrained models can be leveraged, to align the results with more realistic motion patterns. To this end, we adopt a **cross-chunk distribution matching** loss $\mathcal{L}_{DMD}$, which aligns the distribution of generated videos with that of real videos over extended temporal ranges. Specifically, we apply $\mathcal{L}_{DMD}$ to sequences formed by **three adjacent autoregressively generated chunks**. The generator parameters $\theta$ are optimized by minimizing the KL divergence between generated and real distributions:

$$\nabla_\theta \mathcal{L}_{DMD} = \mathbb{E}_t \left( \nabla_\theta \, \text{KL} \left( p_{gen} \, \| \, p_{data} \right) \right), \quad (6)$$

where $p_{gen}$ denotes the distribution of the generated triplet-chunk sequence, $p_{data}$ is the corresponding real distribution learned from the teacher model. Eq. 6, a KL-based regularization loss, is operated in the latent space to minimize the distribution distance. We follow the implementation of previous works (Yin et al., 2024b; Huang et al., 2025).

**Two-Stage Curriculum Training.** AR training is expensive, as each update requires multiple forward passes and additional teacher modules. In VSR, decoding and training at high resolution make this burden prohibitive. We therefore adopt a two-stage curriculum. **Stage I: Initialization** optimizes only $\mathcal{L}_{pix}$ on high-resolution, long clips to fit one-step diffusion, while **Stage II: AR Adaptation** trains at lower resolution with KV-cache and AR inference, adopting both $\mathcal{L}_{pix}$ and $\mathcal{L}_{DMD}$. Empirically, Stage I already yields strong VSR quality, and Stage II converges rapidly to an AR-ready model, allowing us to train the full system at substantially lower cost — using only 4 A800 GPUs.

### 3.4. MovieLQ Dataset and Benchmark

Existing VSR benchmarks, such as UDM10 (Tao et al., 2017), SPMCS (Yi et al., 2019), YouHQ (Zhou et al., 2024), and VideoLQ (Chan et al., 2022b), typically consist of short clips under 100 frames. While suitable for evaluating base quality, they fail to reflect the real-world constraints of long-form video enhancement, where current models must oper-

*Table 2.* VBench Results on UDM10 and MovieLQ.

| Method | UDM10 | | | MovieLQ | | |
|--------|-------|-------|-------|---------|-------|-------|
| | SC | BC | MS | SC | BC | MS |
| UAV | 0.9496 | 0.9489 | 0.9849 | 0.9494 | 0.9456 | 0.9749 |
| MGLD | 0.9413 | 0.9455 | 0.9863 | 0.9432 | 0.9434 | 0.9875 |
| STAR | 0.9450 | 0.9520 | 0.9899 | 0.9546 | **0.9532** | 0.9873 |
| SeedVR | 0.9625 | **0.9536** | 0.9844 | 0.9510 | 0.9405 | 0.9859 |
| Ours | **0.9632** | 0.9523 | **0.9910** | **0.9593** | 0.9513 | **0.9886** |

ate in a memory-efficient, segment-wise manner. To bridge this gap, we introduce **MovieLQ**, a new benchmark comprising 1000-frame-long, single-shot videos sourced from various video hosting sites such as Vimeo and Pixabay with Creative Commons license. Following (Chan et al., 2022b), all clips exhibit real-world degradation patterns without synthetic corruption. To enable fair comparison, we apply progressive patch aggregation following (Wang et al., 2024; 2025b) to current memory-intensive methods.

# 4. Experiments

## 4.1. Experimental Settings

**Datasets.** For training, following (Yang et al., 2024; Liu et al., 2025c) we use the REDS (Nah et al., 2019) dataset and segment the videos into approximately 1K clips. We adopt the RealBasicVSR (Chan et al., 2022b) degradation pipeline to synthesize LQ-HQ pairs. For evaluation, we employ a mix of synthetic and real-world datasets. The synthetic datasets include UDM10 (Tao et al., 2017) and SPMCS (Yi et al., 2019), taking the same degradations as training. For real-world datasets, we apply MVSR4x (Wang et al., 2023b), VideoLQ (Chan et al., 2022b), and our proposed MovieLQ. All experiments are performed under a ×4 upscaling setting.

**Evaluation Metrics.** We employ diverse metrics to thoroughly assess fidelity, perceptual quality, and temporal coherence. For fidelity, we employ full-reference image quality assessment (IQA) metrics including PSNR, SSIM (Wang et al., 2004), LPIPS (Zhang et al., 2018), and DISTS (Ding et al., 2020). For perceptual quality, we adopt no-reference IQA metrics like MUSIQ (Ke et al., 2021) and CLIP-IQA (Wang et al., 2023a), and take DOVER (Wu et al., 2023) for video quality assessment. For temporal consistency, we adopt the flow warping error $E^*_{warp}$ ($\times 10^{-3}$) (Lai et al., 2018) to assess pixel-level consistency. Moreover, we take background consistency (BC), subject consistency (SC), and motion smoothness (MS) from VBench (Huang et al., 2024) to assess temporal consistency at the semantic level, with which a comprehensive evaluation is conducted.

**Implementation Details.** Our InfVSR is built upon the T2V model Wan 2.1 - 1.3B (Wan et al., 2025) and is trained on 4 NVIDIA A800-80G GPUs with AdamW (Loshchilov & Hutter, 2019). We use gradient accumulation for larger

batch size. We set the cache and chunk length both to 3. In Stage 1, we use a resolution of $33 \times 720 \times 1280$ with a window size of $320 \times 480$, a batch size of 8, and a learning rate of $5 \times 10^{-5}$. In Stage 2, the resolution is cropped to $33 \times 480 \times 720$ with a window size of $160 \times 160$, a batch size of 32, and a learning rate of $1 \times 10^{-5}$. The loss scalers we take are $\lambda_{mse} = \lambda_{dists} = \lambda_{temp} = 1$.

## 4.2. Comparison with State-of-the-Art Methods

We compare with SOTA methods: RealBasicVSR (Chan et al., 2022b), RealViFormer (Zhang & Yao, 2024), Upscale-A-Video (Zhou et al., 2024), MGLD-VSR (Yang et al., 2024), STAR (Xie et al., 2025), SeedVR (Wang et al., 2025a), and SeedVR2 (Wang et al., 2026b).

**Quantitative Results.** As shown in Tab 1, InfVSR achieves outstanding performance across diverse datasets. For fidelity, our method frequently ranks top-1 or top-2 on the majority of benchmarks. For perceptual quality, our model obtains the highest scores on five datasets, demonstrating its strong ability to produce visually compelling outputs. In terms of temporal consistency (i.e., $E^*_{warp}$), our approach attains the lowest errors on UDM10 and performs robustly on other sets. Collectively, these findings verify the effectiveness of our model in producing high-fidelity, perceptually natural, and temporally stable videos across both synthetic and real-world scenarios.

**Qualitative Results.** Fig. 3 presents visual comparisons on both synthetic (i.e., SPMCS) and real-world (i.e., VideoLQ) videos. InfVSR effectively handles complex degradations and produces more realistic results. For example, in the first sample, InfVSR successfully recovers the building's structure and textures under severe degradation, whereas the others appear blurry or distorted. Similarly, our method restores clear text edges in the second example. More visual results are provided in the supplementary material.

**Temporal Consistency.** Given the importance of temporal consistency in VSR, we evaluate it from both pixel-level and semantic-level perspectives. We visualize the temporal profile in Fig. 4, where our method exhibits significantly smoother and more coherent transitions over time. Results for VBench metrics are presented in Tab. 2, where our method achieves top-1 or top-2 performance both under full-temporal settings (e.g., UDM10) and progressive patch aggregation pipelines (e.g., MovieLQ). These results highlight the strength of our framework in preserving semantic consistency across long-range sequences. Fig. 5 also illustrates our consistency in identity preserving across chunks with multi-frame comparisons.

**Efficiency.** We compare our method against both multi-step and single-step baselines in terms of runtime (s) and peak memory usage (GB) in Tab. 3. All experiments are conducted on a single NVIDIA A800-80G GPU for fair com-

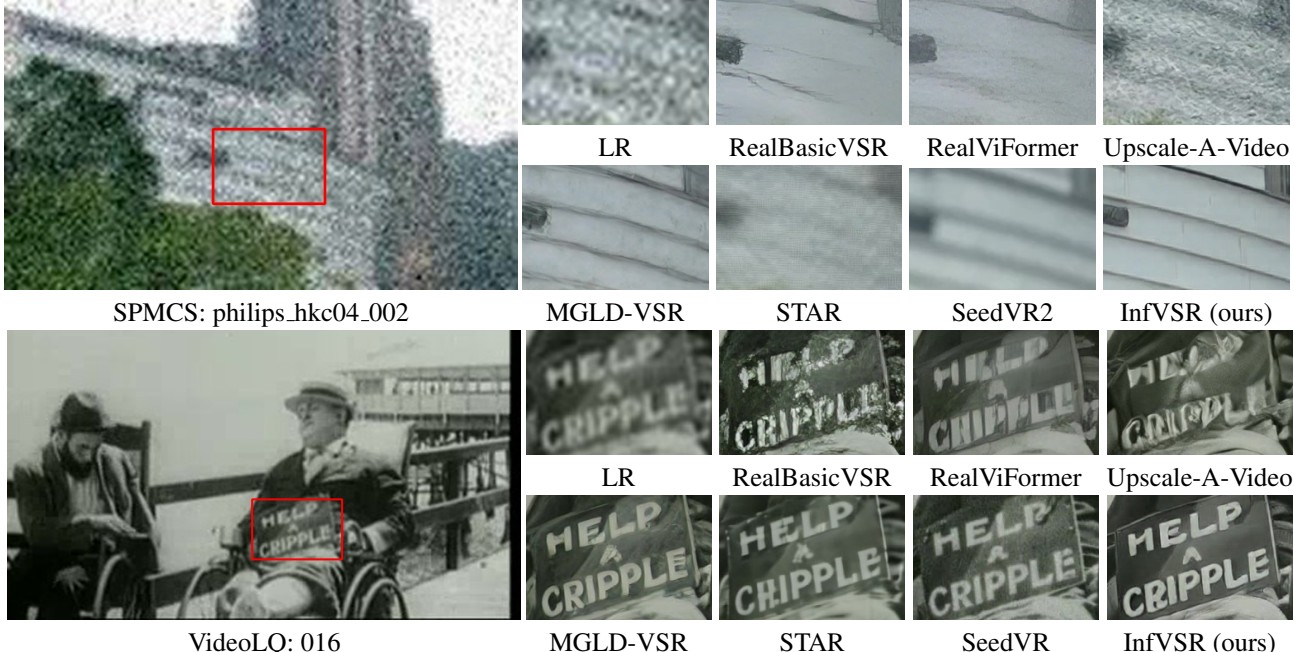

Figure 3. Visual comparison on SPMCS (Yi et al., 2019) and VideoLQ (Chan et al., 2022b).

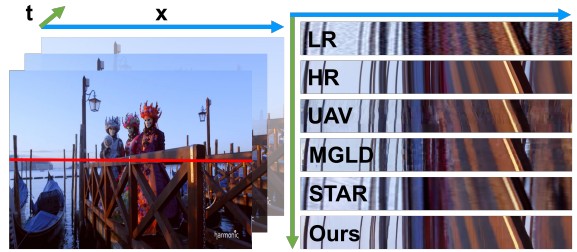

Figure 4. Comparison of temporal profile of SOTA methods (stacking the red line across frames).

Table 3. Comparison of inference step, parameters (B), runtime (s) and memory (GB) of diffusion-based methods.

| Method | Step | Param. | 33×720p Time | 33×720p Mem | 100×720p Time | 100×720p Mem |
|---|---|---|---|---|---|---|
| UAV | 30 | **1.09** | 241.43 | 43.38 | 731.60 | 43.38 |
| MGLD | 50 | 1.57 | 396.06 | 27.70 | 1,200.20 | 27.70 |
| STAR | 15 | 2.49 | 101.59 | 22.14 | 314.84 | 52.99 |
| SeedVR | 50 | 3.40 | 360.66 | 70.44 | 893.03 | 72.44 |
| SeedVR2 | 1 | 3.40 | 37.43 | 61.13 | 68.18 | 61.44 |
| Ours | 1 | 1.41 | **6.82** | **20.39** | **20.70** | **20.39** |

parison. On 720p videos with 33 frames, our method takes only 6.82 seconds, which is 58× faster than the multi-step method MGLD-VSR (Yang et al., 2024) and 5.48× faster than the recent prevailing one-step method, SeedVR2 (Wang et al., 2026b). For long videos (e.g., 100 frames), our memory usage remains constant instead of increasing with the sequence length, and runtime grows linearly with respect to the input length, while still being significantly faster than existing methods. These results demonstrate the high efficiency and scalability of our autoregressive design.

## 4.3. Ablation Study

We conduct comprehensive ablations to evaluate the effectiveness of our proposed designs and settings. All training configurations are kept consistent with settings described in Sec. 4.1 and all experiments are conducted on UDM10 (Tao et al., 2017). Results are presented in Tab. 4.

**Effectiveness of AR Inference.** We verify the effectiveness of our AR inference strategy in Tab. 4a and the right 3 columns in Fig. 5. Compared to simple chunking without cache, AR significantly improves both perceptual quality and temporal consistency, as the latter suffers from small receptive fields and lacks long-range temporal modeling. Although adding overlap and blending (Aggregation) in chunking-based methods improves pixel-level smoothness (as reflected by lower $E_{warp}^*$), it fails to enhance semantic-level consistency. In contrast, our AR design leverages pretrained T2V priors more effectively, leading to better semantic alignment and consistency across long videos.

**Effectiveness of Joint Guidance.** We study the role of joint guidance during inference in Tab. 4b. Without guidance, generation quality degrades across all metrics for lack of semantic information. Extracting separate guidance for each chunk can improve visual quality, but it also introduces inconsistency across segments, resulting in slightly worse performance in $E_{warp}^*$ and semantic consistency metrics. In contrast, our joint guidance strategy reuses a shared visual anchor across adjacent chunks, thereby providing a more stable semantic condition throughout autoregressive inference. As a result, it achieves the best semantic consistency while maintaining competitive perceptual fidelity.

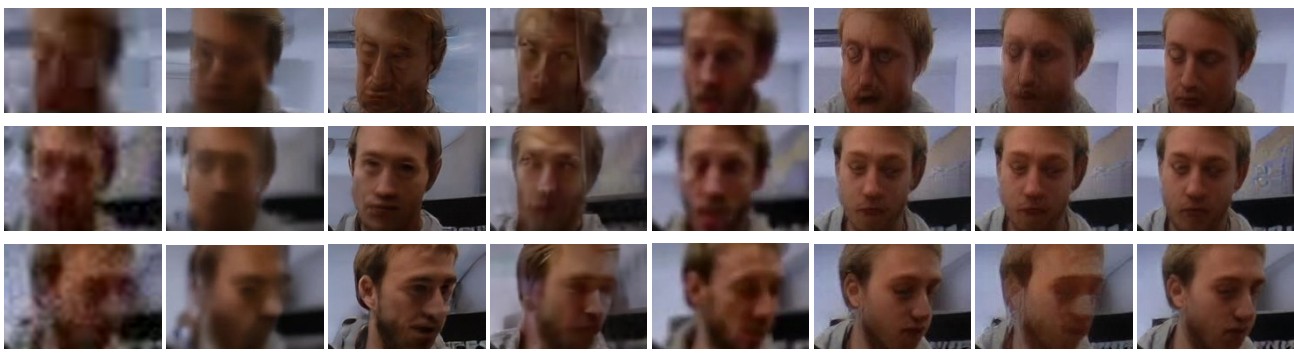

LR          UAV          MGLD-VSR          STAR          SeedVR          Chunking          Aggregation          InfVSR (ours)

*Figure 5.* Multi-frame comparisons for both SOTA comparison and our ablation study on AR Inference.

*(a)* Chunking vs AR.

| Inference | LPIPS | MUSIQ | $E^*_{warp}$ | (BC+SC)/2 |
|---|---|---|---|---|
| (a) Chunking | 0.3178 | 61.29 | 2.20 | 0.9456 |
| (b) Aggregation | 0.3175 | 60.66 | 1.96 | 0.9456 |
| (c) AR (Ours) | **0.2972** | **62.88** | **1.95** | **0.9578** |

*(b)* Guidance.

| Guidance | DISTS | CLIP-IQA | $E^*_{warp}$ | (BC+SC)/2 |
|---|---|---|---|---|
| (a) w/o Guidance | 0.1518 | 0.5128 | 2.01 | 0.9447 |
| (b) Separate | 0.1424 | **0.5165** | 1.97 | 0.9547 |
| (c) Joint (Ours) | **0.1422** | 0.5142 | **1.95** | **0.9578** |

*(c)* AR chunk settings.

| (M, N) | PSNR | LPIPS | CLIP-IQA | $E^*_{warp}$ |
|---|---|---|---|---|
| (a) $(1, 1)$ | 23.79 | 0.3242 | 0.4755 | 2.53 |
| (b) $(5, 5)$ | **24.90** | **0.2963** | 0.4931 | **1.89** |
| (c) $(\infty, 3)$ | 24.73 | 0.2984 | 0.5084 | 2.01 |
| (d) $(3, 3)$ (Ours) | 24.86 | 0.2972 | **0.5142** | 1.95 |

*(d)* Training settings.

| Training | PSNR | LPIPS | CLIP-IQA | $E^*_{warp}$ |
|---|---|---|---|---|
| (a) w/o $\mathcal{L}_{\text{temp}}$ | 24.75 | 0.2972 | **0.5162** | 2.23 |
| (b) w/o Patch | 24.52 | 0.3242 | 0.3997 | 2.03 |
| (c) w/o Stage-I | 24.77 | 0.3125 | 0.3980 | 1.98 |
| (d) Ours | **24.86** | **0.2972** | 0.5142 | **1.95** |

*(e)* DMD loss.

| DMD | PSNR | CLIP-IQA | DOVER | $E^*_{warp}$ | SC | BC |
|---|---|---|---|---|---|---|
| (a) w/o DMD | **25.04** | 0.5028 | 0.7603 | **1.87** | 0.9608 | 0.9483 |
| (b) w/ DMD (Ours) | 24.86 | **0.5142** | **0.7826** | 1.95 | **0.9632** | **0.9523** |

*Table 4.* Ablation study (a–e).

**Influence of Chunk and Cache Size.** In Tab. 4c, we explore different chunk settings (M, N) in AR inference, where M refers to KV-cache length and N refers to the chunk length of latent frames. Using very short chunks like (1, 1) hinders the model from capturing the temporal priors of pretrained T2V models. Increasing the chunk size to (5, 5) brings limited gains over (3, 3) but will nearly triple the quadratic cost of DiT. Moreover, keeping the full KV-cache leads to varying cache lengths at each inference step, making it harder for the model to generalize. Our default choice (3, 3) offers the best trade-off between performance and efficiency.

**Effectiveness of Training Settings.** We ablate key training design choices in Tab. 4d. Removing the temporal loss increases the warping error, showing that explicit frame-difference supervision is useful. Without Stage-I pretraining on high-resolution patches, the model struggles to adapt to long VSR sequences and shows degraded performance. Similarly, removing our proposed pixel-level window loss and training on small datasets, which enables full decoding, also hurts visual quality. This suggests that DiT backbones need proper adaptation to long-sequence VSR.

**Role of DMD Loss.** As shown in Tab. 4e, the introduction of DMD leads to improvements in perceptual quality and semantic consistency. Although the pixel-wise loss alone already enables the model to utilize the pretrained DiT's parameter priors to some extent, adding DMD further improves the results by explicitly encouraging perceptual and semantic alignment. This suggests that DMD helps the model better capture the semantic priors of video diffusion.

## 5. Conclusion

In this work, we propose InfVSR, a scalable and efficient framework for VSR on unbounded-length sequences. By reformulating VSR as an AR-OSD paradigm, our method breaks the length limitations of existing full-sequence approaches. Through causal DiT adaptation with dual-timescale designs and single-step distillation with pixel and distribution matching losses, we enable ultra-efficient inference while preserving temporal coherence. To support evaluation on long videos, we introduce a dedicated benchmark and adopt semantic-level consistency metrics. Extensive experiments demonstrate that our method not only achieves SOTA quality, but also delivers up to 58× speed-up compared to prior multi-step methods such as MGLD-VSR. We believe InfVSR opens new possibilities for long-range video enhancement and lays the foundation for practical deployment of generative VSR systems.

## Impact Statement

This work proposes InfVSR, which enables streaming, long-video super-resolution by reformulating VSR as an autoregressive one-step diffusion process. Like other restoration methods, it could be misused to enhance fabricated or NSFW videos, so deployment should follow licensing or privacy norms and consider provenance safeguards.

## Acknowledgments

This work is supported by the National Natural Science Foundation of China (62501386, 625B2116), CCF-Tencent Rhino-Bird Open Research Fund, CAAI-Tencent Rhino-Bird Open Research Fund, and SJTU Shenlan Program Key Project (SL2023ZD203). This work is also sponsored by AI Hundred Schools Program and is carried out using the Ascend AI technology stack.

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

# Appendix

In the appendix, we provide additional analysis and results. Here is a summary:

# A. Inference Details

## A.1. Inference Latency Analysis

We analyze the runtime cost of each module on 33-frame, 720p video inputs. The breakdown of runtime is shown in Tab. 5 (measured on a single NVIDIA A800-80G GPU). When running on longer videos (e.g., 1000 frames), our memory usage remains **constant**, and latency increases **linearly** with number of frames. It's worth mentioning that the running time of DiT without KV-cache is 1.669s, indicating that the additional computation introduced by the KV cache is less than 11% of the original, which is significantly smaller than even one overlap of a single frame.

*Table 5.* Module-wise runtime breakdown.

| Module | Time (s) |
|---|---|
| VAE encoding | 2.260 |
| Conditioning (with DAPE) | 0.230 |
| DiT denoising (1-step) | 1.844 |
| VAE decoding | 2.493 |
| **Total** | **6.827** |

## A.2. Streaming Inference with Causal VAE

Our streaming strategy aligns with prevailing streaming video generation paradigms (Yin et al., 2025; Huang et al., 2025; Zhang et al., 2025). Here, we further detail the mechanism behind this full-module streaming capability. Unlike conventional methods that require full-sequence buffering, our model enables efficient, streaming-friendly inference by integrating an autoregressive framework with a Causal VAE. Crucially, the VAE employs a cache-based temporal mechanism (maintaining 2 latent frames), allowing it to decode chunks independently without waiting for future context.

During inference, the process operates as a cohesive stream: (1) Inputs are split into overlapping chunks of 3 latent frames (12 pixel frames). (2) Each chunk is super-resolved autoregressively via a rolling KV-cache. (3) Once generated, latents are decoded immediately using the VAE cache.

This design not only ensures low latency and constant memory but also enables the pipeline to be fully parallelized into a three-stage threaded execution (encode, generate, and decode), as illustrated in Fig. 6.

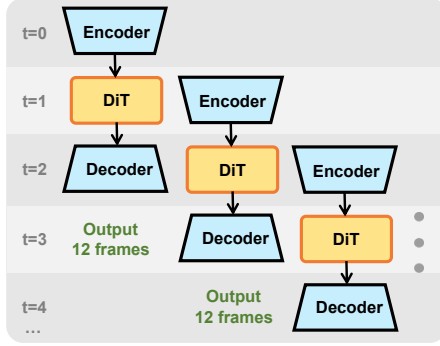

*Figure 6.* Illustration of our streaming pipeline. Users can get results without waiting for the sequence being fully processed.

## A.3. Multi-Shot Processing with PySceneDetect

In our main paper, all experiments are conducted on single-shot videos to ensure a fair comparison, as **existing methods—whether relying on temporal alignment or video diffusion priors—all do not explicitly model scene transitions**, which inevitably leads to information confusion across different scenes. However, as the first method positioned specifically for long-video super-resolution, we adopt a simple yet effective strategy to address this challenge by integrating PySceneDetect (Castellano & contributors). This tool is both highly efficient and accurate. Specifically: (1) When a hard scene cut is detected, we immediately interrupt the cache propagation to prevent temporal confusion between unrelated scenes. (2) For soft transitions, a sensitivity threshold can be employed to trigger an update of the joint visual prompt.

While more sophisticated cross-scene reasoning mechanisms exist, they typically incur high computational costs. Such exploration is left for future work.

## B. Training Details

We detail our Stage-II training procedure in Algorithm 1.

---

**Algorithm 1** InfVSR Stage-II Training Procedure

---

**Input:** Pre-encoded dataset $\mathcal{D} = \{x_{lr}, z_{lr}, x_{gt}\}$, pretrained video diffusion model including denoising network $G_\psi$ and
       VAE decoder $D_\psi$, pretrained ViT embedder $E_{\text{vit}}$, number of video chunks $k$, max KV-cache length $M$, chunk size $N$

**Output:** Trained AR-OSD model $G_\theta$

Initialize student model $G_0$, frozen regularizer $G_\phi$, trainable regularizer $G_{\phi'}$ from $G_\psi$

**while** *training* **do**

     Initialize model output $Z_{sr} \leftarrow [\,]$

     Initialize KV cache KV $\leftarrow [\,]$

     ; // Encode a randomly sampled reference frame

     $x_r \sim \text{Uniform}(x_{lr})$

     $e_r \leftarrow E_{\text{vit}}(x_r)$

     **for** $i = 0$ **to** $k$ **do**

         $z_{lr}^i \leftarrow z_{lr}[iN : (i+1)N]$

         $z_{sr}^i \leftarrow G_\theta(z_{lr}^i, e_r, \text{KV})$

         Append $z_{sr}^i$ to $Z_{sr}$

         Append $\text{kv}^i$ to KV

         **if** $len(KV) > M$ **then**

            Keep only the latest $M$ entries in KV

         **end**

     **end**

     ; // Patch-wise pixel supervision

     $\hat{Z}_{sr} \leftarrow C_{\text{lat}}(Z_{sr})$

     $\hat{x}_{sr} \leftarrow D_\psi(\hat{Z}_{sr})$

     $\hat{x}_{gt} \leftarrow C_{\text{pix}}(x_{gt})$ ;                                      // see Eq. (3)

     $\mathcal{L}_{\text{pix}} \leftarrow \mathcal{L}_{\text{fidel}}(\hat{x}_{sr}, \hat{x}_{gt}) + \mathcal{L}_{\text{temp}}(\hat{x}_{sr}, \hat{x}_{gt})$ ;        // see Eq. (4), (5)

     ; // Distribution Matching

     Sample $t \sim \mathcal{U}\{20, \dots, 980\}$

     Sample $\epsilon \sim \mathcal{N}(0, \mathbf{I})$

     $\hat{z}_t \leftarrow \alpha_t \cdot \text{stopgrad}(\hat{Z}_{sr}) + \sigma_t \cdot \epsilon$

     $z_\phi \leftarrow \text{stopgrad}(G_\phi(\hat{z}_t; c_y))$

     $z_{\phi'} \leftarrow \text{stopgrad}(G_{\phi'}(\hat{z}_t; c_y))$

     $\omega \leftarrow 1/\text{mean}(\|z_\phi - z_{\phi'}\|)$

     $\nabla\mathcal{L}_{\text{DMD}} \leftarrow [\omega(z_{\phi'} - z_\phi)]\frac{\partial \hat{Z}_{sr}}{\partial \theta}$

     ; // Finetune regularizer

     Sample $t \sim \mathcal{U}\{20, \dots, 980\}$

     Sample $\epsilon \sim \mathcal{N}(0, \mathbf{I})$

     $z_t \leftarrow \alpha_t \cdot \text{stopgrad}(\hat{Z}_{sr}) + \sigma_t \cdot \epsilon$

     $\mathcal{L}_{\text{diff}} \leftarrow \mathcal{L}_{\text{MSE}}(G_{\phi'}(z_t; t, c_y), \epsilon)$

     ; // Network Parameter Update

     Update $\theta$ with $\mathcal{L}_{\text{pix}} + \mathcal{L}_{\text{DMD}}$

     Update $\phi'$ with $\mathcal{L}_{\text{diff}}$

**end**

---

# C. Benchmark Details

## C.1. MovieLQ Dataset

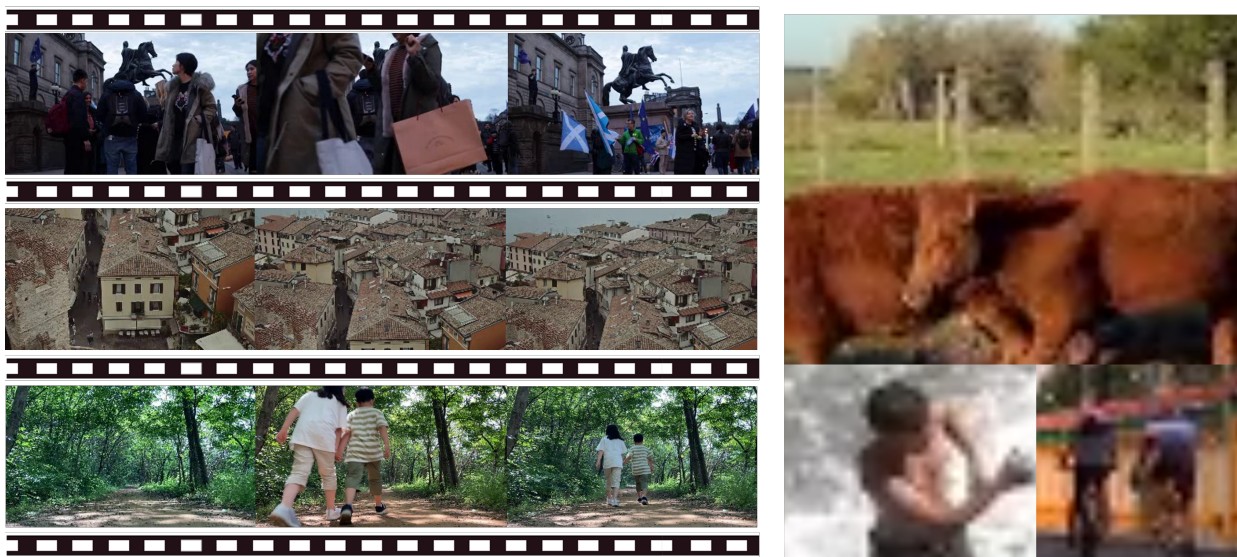

*Figure 7.* Visualization of MovieLQ dataset. It features rare dynamic 1000-frame-long videos (left) and real-world degradations (right).

To remedy the lack of long-sequence VSR benchmarks, we propose **MovieLQ**. Our data collection process largely follows that of VideoLQ (Chan et al., 2022b). We manually collect 10 1000-frame-long, single-shot videos from various video hosting platforms such as Vimeo and Pixabay, all under Creative Commons licenses. The videos are from real-world sources rather than synthetically degraded content. As shown in Fig. 7, MovieLQ exhibits diverse motion patterns and complex real-world degradations.

## C.2. VBench Metric

To compensate for the limited ability of $E^*_{warp}$ to fully characterize temporal consistency, we additionally adopt three temporal-quality metrics from VBench (Huang et al., 2024): **Subject Consistency (SC), Background Consistency (BC), and Motion Smoothness (MS)**. We detail their design and implementation below.

1. **Subject Consistency (SC)** employs DINO features $(d_t)$ for their robust object-level alignment. The score averages the cosine similarity of the current frame with both the first $(d_1)$ and previous $(d_{t-1})$ frames:

$$SC = \frac{1}{T-1} \sum_{t=2}^{T} \frac{1}{2} \left( \langle d_1, d_t \rangle + \langle d_{t-1}, d_t \rangle \right). \tag{7}$$

2. **Background Consistency (BC)** utilizes CLIP embeddings $(c_t)$ to capture global semantic stability. Following the same formulation as SC, it calculates:

$$BC = \frac{1}{T-1} \sum_{t=2}^{T} \frac{1}{2} \left( \langle c_1, c_t \rangle + \langle c_{t-1}, c_t \rangle \right). \tag{8}$$

3. **Motion Smoothness (MS)** validates physical plausibility using RIFE (Huang et al., 2022), based on the premise that realistic motion is predictable. It measures the reconstruction error of interpolated intermediate frames:

$$MS = 1 - \frac{1}{N_{val}} \sum_{t} \mathcal{D}(I_t, \text{RIFE}(I_{t-1}, I_{t+1})), \tag{9}$$

where $\mathcal{D}$ denotes the perceptual difference between the real frame $I_t$ and the interpolated estimation.

# D. More Ablation Studies

**Effect of $\mathcal{L}_{temp}$.** To validate our temporal constraint design, we compare our proposed local temporal loss ($\mathcal{L}_{temp}$) against a representative optical flow-based alignment loss ($\mathcal{L}_{opt} = \|O_n^{HQ} - O_n^{GT}\|_1 = \|F(I_n^{HQ}, I_{n+1}^{HQ}) - F(I_n^{GT}, I_{n+1}^{GT})\|_1$ (Sun et al., 2025)). As shown in Table 6, $\mathcal{L}_{opt}$ fails to improve stability (higher Ewarp) and slightly degrades perceptual qual-

*Table 6.* Ablation study on temporal consistency objectives evaluated on UDM10. **Bold** indicates the best performance.

| Components | PSNR | LPIPS | CLIPIQA | DOVER | Ewarp ($\downarrow$) |
|---|---|---|---|---|---|
| $\mathcal{L}_{fid}$ | 24.75 | 0.2972 | **0.5162** | 0.7794 | 2.23 |
| $\mathcal{L}_{fid} + \mathcal{L}_{opt}$ | 24.50 | 0.2988 | 0.5084 | 0.7753 | 2.24 |
| $\mathcal{L}_{fid} + \mathcal{L}_{temp}$ (Ours) | **24.86** | **0.2972** | 0.5142 | **0.7826** | **1.95** |

ity. We attribute this to our patch-based training strategy: large motions cause pixels to exit the small training windows, rendering flow-based alignment unreliable. In contrast, our $\mathcal{L}_{temp}$ significantly reduces warping error while preserving visual fidelity. This confirms that regularizing temporal variation—rather than enforcing strict warping constraints—is more robust for generative VSR, as it remains compatible with high-fidelity reconstruction even under complex motions.

**Size of the KV-cache.** We further investigate the impact of cache capacity by varying the KV-cache length while keeping the chunk size fixed at 3. A fixed-length cache is crucial for bridging the train–test gap and preventing unbounded memory growth. As shown in Table 7, the absence of a cache (Length 0) results in the poorest performance, highlighting the necessity of temporal context. Increasing the length to 1 brings significant improvements. Our default setting (Length 3) achieves the

*Table 7.* Ablation study on KV-cache length evaluated on MVSR4x. **Bold** indicates the best performance.

| Length | PSNR | LPIPS | CLIPIQA | DOVER | Ewarp ($\downarrow$) |
|---|---|---|---|---|---|
| 0 | 22.35 | 0.3527 | 0.4749 | 0.6571 | 1.28 |
| 1 | 22.43 | 0.3463 | **0.5229** | 0.6861 | 1.09 |
| 3 (Ours) | 22.49 | 0.3452 | **0.5229** | **0.6872** | **1.03** |
| 5 | **22.56** | **0.3450** | 0.5164 | 0.6792 | **1.03** |

optimal balance, securing the best perceptual scores and temporal stability. While extending the length to 5 offers marginal gains in PSNR, it does not further improve perceptual quality, confirming that a length of 3 is sufficient to capture relevant local temporal dependencies for VSR efficiently.

**Efficiency comparison with full attention.** We validate the efficiency of our causal design by comparing it against a full-attention mechanism. As reported in Table 8, our causal design keeps memory usage constant and ensures runtime grows linearly with the number of frames. Specifically, we can process 1000 frames (f1k) with steady memory consumption (e.g., 14.3GB

*Table 8.* Efficiency comparison between our causal design and full attention. **Bold** indicates the best performance.

| Metric | Causal-f1000(ours) | Full-f33 | Full-f100 | Full-f200 | Full-f300 |
|---|---|---|---|---|---|
| 720p fps | **26.67** | 19.54 | 10.57 | 5.84 | - |
| 720p Mem(GB) | 14.3 | 12.6 | 27.7 | 51.7 | OOM |
| 1080p fps | **7.73** | 5.07 | 2.34 | - | - |
| 1080p Mem(GB) | 27.9 | 23.4 | 60.0 | OOM | OOM |

for 720p). In contrast, the full-attention baseline leads to significant memory and computation growth as the sequence length increases, eventually resulting in Out-Of-Memory (OOM) errors at f200 or f300. This confirms that our causal architecture brings substantial efficiency improvements, making it feasible for long-video super-resolution.

**Discussion on error accumulation.** We further analyze whether autoregressive inference causes error accumulation over long sequences. Empirically, we split each 1000-frame video in MovieLQ into five consecutive temporal ranges and report the DOVER score for each range. As shown in Table 9, our method consistently outperforms the compared methods across all segments, and its performance does not show a monotonic decline over time. This indicates that our autoregressive inference does not introduce ob-

*Table 9.* Length-vs.-performance analysis on MovieLQ. **Bold** indicates the best performance.

| DOVER | 1–200 | 201–400 | 401–600 | 601–800 | 801–1000 |
|---|---|---|---|---|---|
| UAV | 0.7963 | 0.8028 | 0.7879 | 0.7858 | 0.7913 |
| SeedVR2 | 0.8101 | 0.8056 | 0.7917 | 0.7986 | 0.8022 |
| Ours | **0.8608** | **0.8594** | **0.8493** | **0.8532** | **0.8586** |

servable long-term degradation. Theoretically, error accumulation is less severe in VSR than in open-ended autoregressive video generation. In video generation, each new segment is synthesized mainly from previously generated content, so errors may be recursively propagated. In contrast, VSR is a conditional restoration task, where each chunk is directly anchored by its corresponding LR observations. These LR inputs continuously constrain the structure, motion, and content of the output, while the cache mainly serves as auxiliary temporal context rather than the primary source of generation. Therefore, our framework benefits from autoregressive temporal modeling while largely avoiding severe error accumulation.

## E. Comparison with Concurrent Streaming VSR Works

Before the submission, we further notice two concurrent streaming VSR frameworks: Stream-DiffVSR (Shiu et al., 2025) and FlashVSR (Zhuang et al., 2025). Since Stream-DiffVSR does not target real-world degradations, we focus our comparison on FlashVSR. Additionally, we evaluate our method against DLoraL (Sun et al., 2025), a window-based approach that operates in a quasi-streaming manner. Compared with these methods, **our approach achieves competitive visual quality while leading in fidelity and temporal consistency, fully validating the effectiveness of our consistency-driven strategy.**

*Table 10.* Quantitative comparison on UDM10 and VideoLQ.

| Datasets | Metrics | DLoraL | FlashVSR | Ours |
|---|---|---|---|---|
| UDM10 | PSNR ↑ | 24.44 | 23.89 | 24.86 |
| | SSIM ↑ | 0.7104 | 0.7056 | 0.7274 |
| | LPIPS ↓ | 0.3284 | 0.2809 | 0.2972 |
| | DISTS ↓ | 0.1947 | 0.1447 | 0.1422 |
| | MUSIQ ↑ | 63.65 | 63.58 | 62.88 |
| | CLIP-IQA ↑ | 0.6555 | 0.4749 | 0.5142 |
| | DOVER ↑ | 0.7219 | 0.7953 | 0.7826 |
| | $E^*_{warp}$ ↓ | 3.40 | 2.47 | 1.95 |
| VideoLQ | MUSIQ ↑ | 60.46 | 54.93 | 56.26 |
| | CLIP-IQA ↑ | 0.5942 | 0.3930 | 0.4454 |
| | DOVER ↑ | 0.7469 | 0.7845 | 0.7556 |
| | $E^*_{warp}$ ↓ | 9.41 | 12.06 | 7.52 |

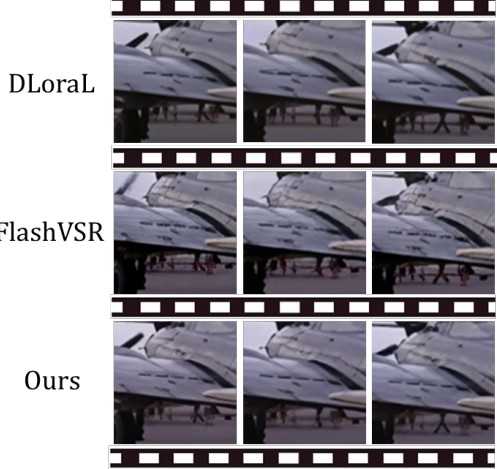

*Figure 8.* Visual comparison of the methods.

## F. Limitations

While our work successfully realizes consistency-driven streaming generative VSR, demonstrating superior performance across multiple metrics and visual assessments, we acknowledge certain limitations and avenues for improvement:

**First, regarding training scale.** Constrained by computational resources, we could not provide results based on massive-scale training resources, larger dataset scales or higher training resolutions. Our current setup (4 GPUs, 1k videos) highlights the data efficiency of our method and its ability to effectively leverage strong video pre-training priors. However, we acknowledge that typically, scaling up training yields stronger generalization capabilities and would enable computationally expensive yet performance-enhancing techniques, such as progressive distillation (Wang et al., 2026b; Zhuang et al., 2025).

**Second, regarding architectural redundancy.** Our research focused on the design of the streaming mechanism and single-step diffusion training strategies, rather than purely redundancy reduction. Our framework is theoretically orthogonal to compression techniques such as quantization, pruning, and sparsity; the combination of these efficient architectures with our method remains a promising direction for future exploration.

**Finally, regarding model priors.** Although we successfully harnessed strong pre-trained priors to achieve state-of-the-art performance, we must acknowledge the intrinsic upper bounds of these priors. These include the reconstruction limits and compression errors inherent in the VAE, as well as the capacity ceilings of the underlying DiT backbone.

In the future, we plan to explore larger training scales, integrate our method with more efficient model architectures, and investigate strategies to transcend the current limits of diffusion priors.

## G. More Visual Results

We provide more visual comparisons in Fig. 9 and Fig. 10.

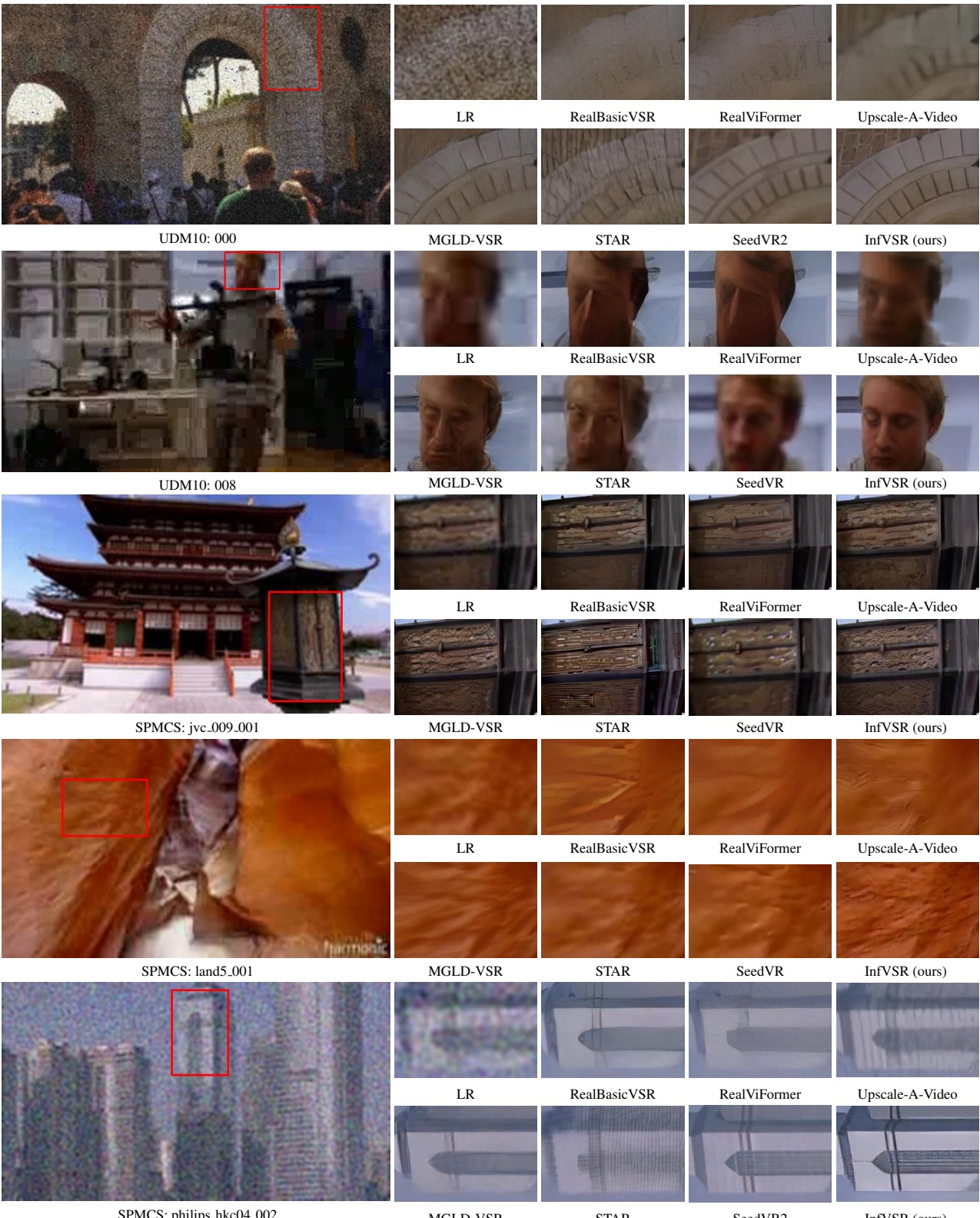

*Figure 9.* Visual comparison on synthetic datasets.

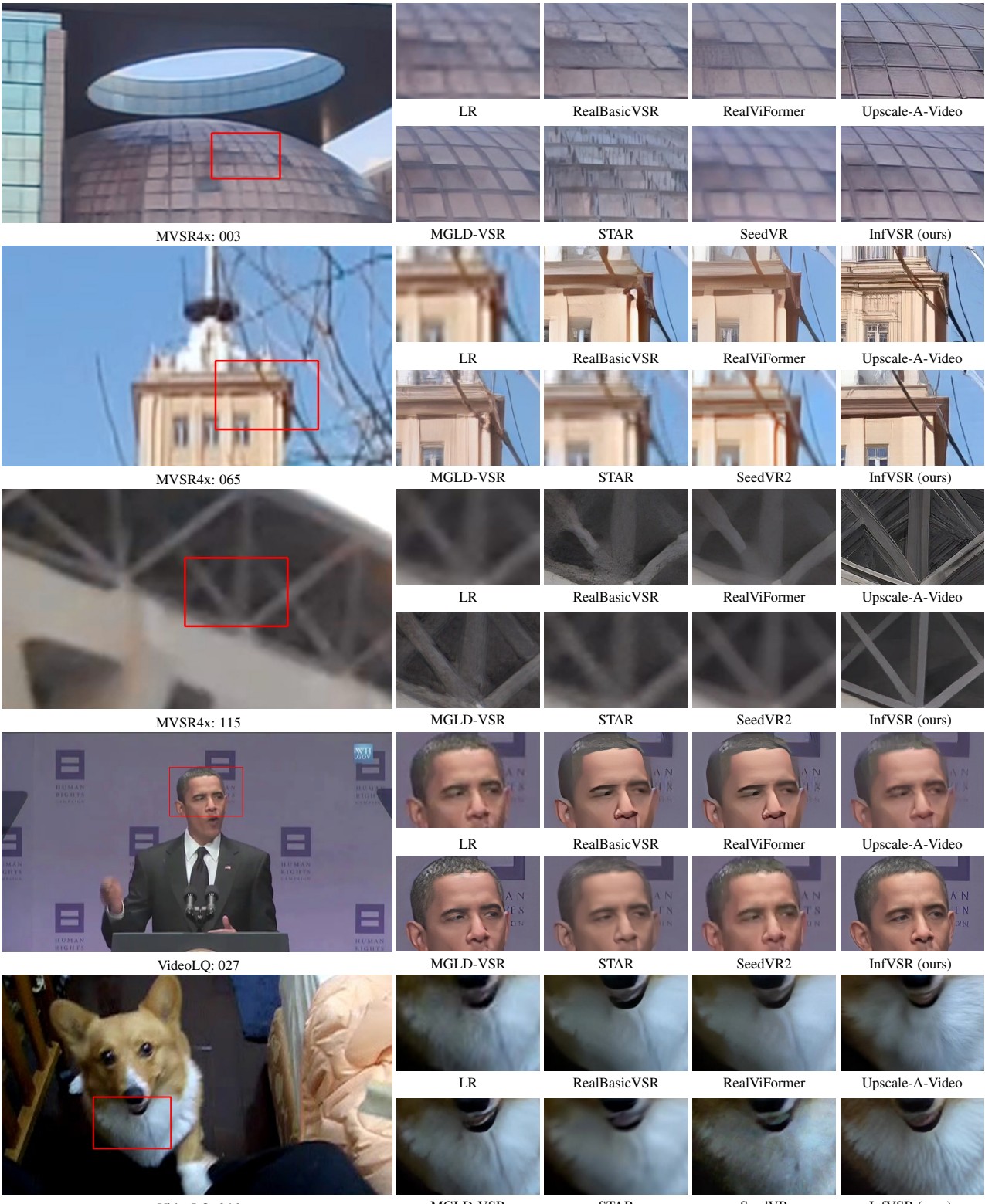

*Figure 10.* Visual comparison real-world datasets.

