# OpenReview forum: "InfVSR: Toward Consistency-Driven Streaming Generative Video Super-Resolution"
_ICML.cc/2026/Conference — ICML 2026 regular_

### Official Review · Reviewer_WGFo · 2026-02-26

**Soundness:** 3
**Presentation:** 3
**Significance:** 3
**Originality:** 3
**Overall Recommendation:** 5
**Confidence:** 5

**Summary:**

This paper proposes InfVSR, a consistency-driven streaming VSR framework. For modeling consistency, the authors reformulate VSR as an autoregressive one-step-diffusion paradigm, utilizing rolling KV-cache and joint visual guidance to maintain both local transitions and global semantic coherence. To enforce consistency during training, the framework employs patch-wise pixel supervision to ensure spatial stability under memory constraints. Furthermore, the paper introduces MovieLQ, a new benchmark featuring 1000-frame real-world degraded videos, and adopts VBench metrics to evaluate semantic-level temporal stability more effectively. Experimental results demonstrates state-of-the-art performance in both visual quality and computational efficiency.

**Compliance With Llm Reviewing Policy:**

Affirmed.

**Final Justification:**

My concerns have been addressed. So I will raise my score.

**Key Questions For Authors:**

- It is noticed that the authors solely rely on a sliding window mechanism to update the cache and only ablate the impact of different cache lengths. Existing works have explored the importance decay of historical information[1], or developed specific modules to process historical features[2]. Could the authors compare the impact of other cache updating or management mechanisms on VSR performance?

- DOVE[3] is a relatively recent and influential work in this domain, yet it is notably absent from the compared baselines. Could the authors provide a comparison with DOVE?

[1] Zhang, Lvmin, and Maneesh Agrawala. "Packing input frame context in next-frame prediction models for video generation." arXiv e-prints (2025): arXiv-2504.

[2] Zhang, Lvmin, et al. "Pretraining Frame Preservation in Autoregressive Video Memory Compression." arXiv preprint arXiv:2512.23851 (2025).

[3] Chen, Zheng, et al. "Dove: Efficient one-step diffusion model for real-world video super-resolution." arXiv preprint arXiv:2505.16239 (2025).

**Limitations:**

Yes

**Strengths And Weaknesses:**

# Strengths

+ The evaluation is comprehensive and detailed, covering diverse quality, temporal, and efficiency metrics (parameters and runtimes).

+ Compared to existing methods, the training of this work can be performed on 4 GPUs and only requires the REDS dataset, which is training efficient.

+ The newly adopted metrics can reveal some interesting findings. For instance, while DMD increases the warping error due to generating sharper textures, it actually improves semantic coherence.



# Weaknesses

- The paper contains a few typos and formatting issues. For instance, the numbers are missing in the rightmost legend of Fig. 1, and "AAR" in Table 4\(c\) should be corrected to "AR".

- In the ablation study, it is observed that a chunk length of 5 achieves stronger consistency than the method's default setting of 3. Empirically, causal attention lacks a receptive field for future frames compared to full attention, which inherently weakens its capability for temporal modeling. This seems contradictory to the paper's core motivation of being "consistency-driven." The authors should rejustify their rationale for adopting the current streaming design.

- The proposed method primarily follows the AR generation and DMD training paradigms similar to Self-Forcing[2]. However, recent studies[3][4][5] have demonstrated that such approaches are susceptible to error accumulation and often require specific strategies for mitigation. There is a valid concern that the framework presented in this paper might also suffer from similar risks of error accumulation over long sequences.

- The paper in [6] has shown that the pretrained AR based video generation models tend to have 'dummy heads' where there are some heads attending exclusively to the current frame despite previous frames are available. I wonder whether this observation extends to video restoration filed, and whether the dummy head could make the proposed model more efficient?

[1] Yin, Tianwei, et al. "One-step diffusion with distribution matching distillation." Proceedings of the IEEE/CVF conference on computer vision and pattern recognition. 2024.

[2] Huang, Xun, et al. "Self forcing: Bridging the train-test gap in autoregressive video diffusion." arXiv preprint arXiv:2506.08009 (2025).

[3] Yang, Shuai, et al. "Longlive: Real-time interactive long video generation." arXiv preprint arXiv:2509.22622 (2025).

[4] Liu, Kunhao, et al. "Rolling forcing: Autoregressive long video diffusion in real time." arXiv preprint arXiv:2509.25161 (2025).

[5] Cui, Justin, et al. "Self-forcing++: Towards minute-scale high-quality video generation." arXiv preprint arXiv:2510.02283 (2025).

[6] Hang Guo, et al. "Efficient Autoregressive Video Diffusion with Dummy Head" arXiv preprint arXiv:2601.20499(2026).

---

> ### Author Rebuttal · Authors · 2026-03-31
>
> > `Q4-1` Typos and formatting issues.
>
> `A4-1` Thanks for pointing out. We will correct them in the revision.
>
> > `Q4-2` The authors should rejustify their rationale for adopting the current streaming design.
>
> `A4-2` First, although full attention or a longer chunk length can bring slightly better consistency due to a larger receptive field, they also introduce substantially higher memory and runtime costs. More importantly, their computational cost grows rapidly with sequence length, which makes them **impractical for long videos** and often requires temporal truncation in real-world inference. Such truncation itself breaks temporal continuity. In contrast, our streaming design enables smooth transitions across arbitrarily long videos **without truncation**, which we believe is essential for consistency in practical long-video VSR.
>
> Second, our ablation results show that further increasing the chunk length leads to only **marginal consistency gains**, while the quality of individual frames may even decrease. This suggests an inherent t**rade-off between single-frame fidelity and temporal consistency**. Therefore, the default chunk length of 3 is chosen as the **best overall balance** between efficiency, restoration quality, and temporal consistency.
>
> Therefore, we believe the current streaming design is a well-justified and practically meaningful choice.
>
> > `Q4-3` ...risks of error accumulation over long sequences.
>
> `A4-3` We address this from **both empirical and theoretical perspectives**.
>
> Empirically, our supplementary material includes multiple 1000-frame examples, together with quantitative results, which show **no observable performance degradation over time**.
>
> Theoretically, we believe the risk of error accumulation is **fundamentally weaker in VSR than in autoregressive video generation**. In video generation, each new chunk is synthesized from previously generated content, so prediction errors can be recursively propagated and amplified. In contrast, VSR is a conditional restoration task: for every chunk, the model always receives the corresponding low-resolution frames as direct observations. These LR inputs provide a **strong per-chunk anchor** for structure, motion, and content, which continuously constrains the solution space and prevents the model from freely drifting away. Under this setting, the cache mainly serves as an auxiliary mechanism for temporal information transfer **rather than the primary source of content generation**.
>
> Overall, this suggests no risk of error accumulation.
>
> > `Q4-4` ...whether this observation extends to video restoration filed, and whether the dummy head could make the proposed model more efficient?
>
> `A4-4` We find that the dummy head phenomenon also appears in VSR. Following the algorithm in the cited paper, we separate out **50% dummy heads**, and report results below.
>
> |MVSR4x|PSNR|LPIPS|CLIPIQA|DOVER|Ewarp|DiT Runtime(s/100x720p)|
> |-|-|-|-|-|-|-|
> |w/ dummy forcing|22.49|0.3449|0.5217|0.6771|1.09|5.33|
> |Ours|22.49|0.3452|0.5229|0.6872|1.03|5.67|
>
> Results show that InfVSR is compatible with this acceleration framework: with a 50% pruning ratio, it maintains comparable performance, which further demonstrates our generalization ability. The speedup is relatively limited, however, since KV-cache is not the main source of computation in InfVSR.
>
> > `Q4-5` Could the authors compare the impact of other cache updating or management mechanisms on VSR performance?
>
> `A4-5` We further compare our current rolling strategy with two representative cache update mechanisms: (1) **anchor + rolling update**, which preserves the first frame as a persistent anchor, and (2) a **learnable distance-aware compression strategy** following FramePack.
>
> |MVSR4x|PSNR|LPIPS|CLIPIQA|DOVER|Ewarp|
> |-|-|-|-|-|-|
> |(1)|22.27|0.3501|0.4897|0.6523|1.12|
> |(2)|22.47|0.3459|0.5220|0.6869|1.03|
> |Ours|22.49|0.3452|0.5229|0.6872|1.03|
>
> Results suggest that a fixed anchor may introduces extra interference, while compressing distant frames brings marginal change. Overall, our current rolling update is already sufficient for VSR.
>
> > `Q4-6` Could the authors provide a comparison with DOVE?
>
> `A4-6` Yes, we provide it below.
>
> |UDM10|PSNR|SSIM|LPIPS|DISTS|MUSIQ|CLIPIQA|DOVER|Ewarp|
> |-|-|-|-|-|-|-|-|-|
> |DOVE|26.48|0.7827|0.2696|0.1492|61.68|0.5108|0.7805|1.77|
> |Ours|24.86|0.7274|0.2972|0.1422|62.88|0.5142|0.7826|1.95|
>
> |MVSR4x|PSNR|SSIM|LPIPS|DISTS|MUSIQ|CLIPIQA|DOVER|Ewarp|
> |-|-|-|-|-|-|-|-|-|
> |DOVE|22.42|0.7523|0.3476|0.2363|61.95|0.5453|0.6984|0.78|
> |Ours|22.49|0.7373|0.3452|0.2107|64.03|0.5229|0.6872|1.03|
>
> |Eff.|Params(B)|33x720p(s)|100x720p(s)|
> |-|-|-|-|
> |DOVE|5.65|14.90|59.83|
> |Ours|1.42|6.82|20.70|
>
> Our method achieves performance comparable to DOVE, while being significantly more efficient, which clearly demonstrates its advantage.

---

> > ### Author Rebuttal · Reviewer_WGFo · 2026-04-03
> >
> > Thanks for your rebuttal, my concerns have been addressed. So I will raise my score.

---

> > > ### Author Response · Authors · 2026-04-03
> > >
> > > Thanks for your positive response and support of our paper. We appreciate your time and effort in reviewing our paper, and are grateful for your constructive feedback throughout this process. We will incorporate these valuable results and explanations into the revised paper and continue to refine our work based on your feedback.

---

### Official Review · Reviewer_hLcA · 2026-03-11

**Soundness:** 3
**Presentation:** 3
**Significance:** 3
**Originality:** 2
**Overall Recommendation:** 4
**Confidence:** 2

**Summary:**

This paper addresses two key challenges in generative video super-resolution (VSR): inefficiency due to multi-step denoising and poor temporal consistency caused by independent chunk processing. The authors propose InfVSR, which reformulates VSR as an autoregressive one-step diffusion (AR-OSD) process built on a pretrained text-to-video (T2V) DiT model. The method introduces a rolling KV-cache for local temporal smoothness, joint visual guidance from LR reference frames for global coherence, patch-wise pixel supervision for memory-efficient training, and cross-chunk distribution matching (DMD) for semantic consistency. The authors also introduce MovieLQ, a new 1000-frame long-video benchmark, and adopt semantic-level temporal consistency metrics from VBench. The method claims state-of-the-art quality with up to 58x speed-up over multi-step diffusion baselines.

**Compliance With Llm Reviewing Policy:**

Affirmed.

**Final Justification:**

The authors further clarified the novelty and improved the experimental validation to address my concerns. So, I raised my score to weak accept.

**Key Questions For Authors:**

Please see the Limitations part for the Questions.

**Limitations:**

1. The rolling KV-cache mechanism is directly borrowed from LLM-style autoregressive inference and from concurrent streaming video generation works. The cross-chunk distribution matching loss is taken from DMD (Sun et al., 2024) and self-forcing (Huang et al., 2025). The patch-wise pixel supervision is a standard random cropping strategy. The joint visual guidance via DAPE-encoded reference frames injected into cross-attention is similar to IP-Adapter-style conditioning. While the combination is reasonable, the paper does not sufficiently discuss what is truly new versus assembled from existing techniques.

2.  The claim of being "the first T2V-based autoregressive-one-step-diffusion framework for real-world VSR" is potentially overclaiming. Works like SeedVR2 (Wang et al., 2025a[a]) also perform one-step video restoration with DiT backbones, and Stream-DiffVSR (Shiu et al., 2025) and FlashVSR (Zhuang et al., 2025) also target streaming VSR with autoregressive diffusion. The paper acknowledges these in Sec. E of the supplement but only after the main claims are made, which is misleading.

[a] Wang, Jianyi, et al. "Seedvr2: One-step video restoration via diffusion adversarial post-training." arxiv 2025


3. The MovieLQ benchmark, while a useful contribution, is very small (only 10 videos). This severely limits its statistical reliability for benchmarking. Variance across runs or across video selection could significantly affect rankings. The paper does not report confidence intervals or standard deviations on any benchmark.

4. Temporal consistency evaluation has significant gaps. The E*_warp metric relies on optical flow estimation, which is itself noisy and unreliable on real-world degraded videos. The VBench metrics (SC, BC, MS) are reported only on UDM10 and MovieLQ (Table 2), not on all five benchmarks. The paper does not include any user study or perceptual evaluation by human raters, which is critical for assessing temporal consistency in video, a fundamentally perceptual phenomenon.

5. The paper evaluates only at 4x upscaling on 720p output. There is no evaluation at higher scales (e.g., 8x) or higher resolutions (e.g., 1080p, 4K), which limits the understanding of how well the method generalizes. The efficiency comparison in Table 3 also only covers 720p, though the supplementary mentions 1080p briefly.

6. The training data is relatively small (REDS dataset, ~1K clips) and the training is done on only 4 A800 GPUs. The paper acknowledges this limitation but does not adequately discuss how this constrains the method's generalization capability compared to baselines like SeedVR/SeedVR2 that are trained on much larger datasets. This makes it unclear whether the performance gains come from the method itself or from the T2V pretrained prior.


7. The single-shot video assumption is a significant practical limitation. The multi-shot handling via PySceneDetect (described only in the supplementary) is a simple heuristic that interrupts cache propagation at scene boundaries. This means the method's core consistency mechanism breaks down at every scene change, which is extremely common in real-world video content. This limitation deserves more prominent discussion.


8. The paper lacks analysis of failure cases. When does the method produce artifacts? How does it handle extreme motions, occlusions, or appearance changes? The supplementary mentions limitations regarding training scale and model priors but provides no concrete failure examples.



9. The PSNR numbers are generally not the best among compared methods (e.g., on UDM10, RealViFormer achieves 24.64 vs. InfVSR's 24.86 which is close, but on SPMCS, RealViFormer gets 22.72 vs. 22.25). The method's strength lies primarily in perceptual metrics (MUSIQ, CLIP-IQA, DOVER), which is expected given the generative diffusion backbone. The paper should more honestly characterize this trade-off rather than claiming broad superiority.


10. The cross-chunk DMD loss is applied over "three autoregressively generated chunks" during training, but inference may involve hundreds of chunks. There is no analysis of whether distribution drift accumulates over very long sequences beyond the training horizon, despite this being a core motivation of the paper.

**Strengths And Weaknesses:**

1. The AR-OSD formulation is a well-motivated formulation that naturally addresses the streaming and consistency requirements of long-form VSR.
2. The paper provides a thorough experimental evaluation across five benchmarks (UDM10, SPMCS, MVSR4x, VideoLQ, MovieLQ) with a comprehensive set of metrics covering fidelity (PSNR, SSIM, LPIPS, DISTS), perceptual quality (MUSIQ, CLIP-IQA, DOVER), and temporal consistency (E*_warp, VBench SC/BC/MS).

Please see the Limitations part for the weaknesses.

---

> ### Author Rebuttal · Authors · 2026-03-31
>
> > `Q3-1` The paper does not sufficiently discuss what is truly new versus assembled from existing techniques.
>
> `A3-1` Thanks for your concern. Please refer to `A2-1` for our novelty rejustification.
>
> > `Q3-2` The claim of being "the first..." is misleading.
>
> `A3-2` Thanks for pointing out. We will revise the main paper accordingly.
>
> > `Q3-3` The MovieLQ benchmark is very small.
>
> `A3-3` We need to clarify that the choice of 10 videos is made after careful consideration, because we need to keep the testing and metric-computation cost within a practical range as other benchmarks. As shown below, 10 videos is a common scale for current mainstream VSR benchmarks, and ours already has the largest total number of frames.
>
> ||UDM10|SPMCS|MVSR4x|VideoLQ|MovieLQ(Ours)|
> |-|-|-|-|-|-|
> |Video|10|30|15|50|10|
> |Frame|320|900|1500|5000|10000|
>
> We further clarify the evaluation fairness: **the dataset was fixed before any metric testing was conducted**. All methods were evaluated using the **same LR inputs** and the **default / commonly used hyperparameter settings** from their open-source code. Therefore, the influence of randomness across different runs is very limited, and we do not consider variance / standard deviation necessary. This is also consistent with the common practice of existing VSR methods.
>
> > `Q3-4` Temporal consistency evaluation has significant gaps.
>
> `A3-4` We agree Ewarp has inherent limitations, and that's why we adopt multiple complementary metrics for a more comprehensive evaluation, as discussed in `A1-1`.
>
> We further supplement more VBench results below.
>
> |SPMCS|BC|SC|MS|
> |-|-|-|-|
> |UAV|0.9541|0.9672|0.9817|
> |MGLD|0.9726|0.9749|0.9890|
> |SeedVR|0.9805|0.9771|0.9894|
> |Ours|0.9716|0.9805|0.9912|
>
> |MVSR4x|BC|SC|MS|
> |-|-|-|-|
> |UAV|0.9518|0.9435|0.9811|
> |STAR|0.9678|0.9514|0.9915|
> |SeedVR2|0.9561|0.9548|0.9889|
> |Ours|0.9514|0.9557|0.9911|
>
> We additionally conduct a user study using the GSB test, following SeedVR2. Specifically, we randomly sample 30 videos from all 5 test sets and employ 10 raters with computer vision backgrounds to score them.
>
> ||Fidelity|Quality|Overall|
> |-|-|-|-|
> |UAV|-33.3%|-46.7%|-36.7%|
> |MGLD|-20.0%|-36.7%|-33.3%|
> |STAR|-10.0%|-30.0%|-26.7%|
> |SeedVR2|13.3%|-20.0%|-16.7%|
> |Ours|0.0%|0.0%|0.0%|
>
> > `Q3-5` The paper evaluates only at 4x upscaling on 720p output.
>
> `A3-5` First, diffusion-based VSR methods are typically trained and compared under the 4× setting, so 8× evaluation is not standard. Second, our evaluations are not limited to 720p(e.g., MVSR 1024x1024, VideoLQ 2K/3K).
>
> > `Q3-6` ...unclear whether the performance gains come from the method itself or from the T2V pretrained prior.
>
> `A3-6` We believe that **effectively and properly leveraging the pretrained T2V generative prior** is itself a key strength of our method.  Our proposed design provides meaningful insight into how such a prior can be used for streaming VSR without disrupting its generative capability.
>
> In addition, our ablation studies show consistent improvements brought by both the model design and the training components, indicating that the gains are **not merely inherited from the pretrained model**, but also come from the proposed framework itself.
>
> > `Q3-7` The single-shot video assumption is a significant practical limitation.
>
> `A3-7` First, we need to clarify that single-shot video is **never** a narrow corner case. In many real-world applications, such as surveillance, livestreaming, and driving videos, temporally continuous segments can span thousands of frames or even hours. Effectively solving long-horizon consistency in this setting is therefore **already a substantial practical contribution**.
>
> Then for scene changes, resetting the cache is **not** a breakdown, but a principled choice. Scene cuts are incompatible with the temporal compression of pretrained T2V LDMs; forcing cache propagation across shot boundaries would only lead to information confusion.
>
> > `Q3-8` The paper lacks analysis of failure cases.
>
> `A3-8` Like other OSD methods, our method still struggles to actively generate tiny text or faithfully recover severely corrupted fine details, since our model is trained under a fidelity constraint.
>
> For extreme motion, our method is in fact relatively more robust: it maintains full 3D global attention within each 24-frame window, instead of relying on explicit alignment or optical flow, which can fail under large displacement or occlusion.
>
> > `Q3-9`  The paper should more honestly characterize this trade-off rather than claiming broad superiority.
>
> `A3-9` We will revise the main paper accordingly, while noting that we do not claim the best result on every metric; rather, our method reaches a favorable point on the quality–efficiency Pareto frontier, which we believe is meaningful.
>
> > `Q3-10` There is no analysis of whether distribution drift accumulates over very long sequences.
>
> `A3-10` Please refer to `A4-3` for explanation and `A2-4` for experimental results.

---

> > ### Author Rebuttal · Reviewer_hLcA · 2026-04-03
> >
> > I appreciate the authors’ effort in providing additional experiments and clarifications during the rebuttal. However, I remain unconvinced that the core concerns have been adequately addressed.
> >
> > First, the authors’ responses do not clearly resolve the question of novelty. While they argue that the contribution lies in the integration of existing components, it remains unclear what fundamentally new insight or capability is introduced beyond prior work. The method still appears largely as an engineering combination of known techniques.
> >
> > Second, several key claims are still insufficiently supported by rigorous evaluation. In particular, concerns regarding evaluation at higher scales (e.g., 8x), failure cases.
> >
> > Third, it remains unclear to what extent the performance gains are driven by the pretrained T2V prior, as opposed to the proposed method itself, and this is not convincingly validated.

---

> > > ### Author Response · Authors · 2026-04-04
> > >
> > > We sincerely thank you for your timely feedback, and are glad to see that most of the initial 10 questions have been addressed. For the remaining concerns, we address them with more analyses and results.
> > >
> > > > `Q3-11` The authors’ responses do not clearly resolve the question of novelty
> > >
> > >  `A3-11` Thank you for the concern. We respectfully disagree that our method is merely an engineering combination. For each technique in your`Q3-1`, we first directly clarify the key differences and our innovations below.
> > >
> > > ||Technique|Related Prior Art|Our Distinctive Design|Resulting New Capability|
> > > |-|-|-|-|-|
> > > |1|rolling KV-cache|LLM/streaming vidgen|fully consider VSR's nature with LR inputs and carefully validate chunking/cache strategy|outperform anchor-frame/compression-based cache update methods (`A4-5`) and adaptive chunking strategies (`A2-2`); well-balance performance and efficiency (Main Tab4c, Supp. Sec D)|
> > > |2|cross-chunk DMD|DMD/selfforcing|DMD is not autoregressive. Self-forcing is multi-step and requires an **additional forward pass to extract clean cache**. In contrast, our method completes super-resolution and cache update in **only one forward pass**.|a highly efficient AR-OSD training scheme, rather than directly inheriting heavier prior pipelines|
> > > |3|patch-wise pixel supervision|random cropping|not ordinary random patch cropping, but an asymmetric design: the DiT keeps the uncropped global processing, while only the decoder part is cropped|enables training the DiT at high resolution while greatly reducing memory usage, improving performance (Main Tab4d)|
> > > |4|joint visual guidance|IP-Adapter-style conditioning|not strong control, but soft semantic prompting; further innovatively design a cross-chunk sharing mechanism|replace heavy VLMs; reduce the number of feature extraction calls and improve cross-chunk consistency (`A1-8`)|
> > >
> > > Beyond these techniques, our work contains more innovations, such as formulation, evaluation and benchmark. For their underlying insight and detailed discussion, we respectfully refer you to `A1-7`.
> > >
> > > Finally, contribution and novelty are not defined only by conceptual newness, as reflected in the ICML announcement (`A1-7`). And as recognized by **Reviewer JuG6** (“one of the first attempts at a complete system design for the practical setting: streaming generative VSR”) and **Reviewer WGFo** (“reveals interesting findings”), our work provides both methodological innovation and system-level contribution. We therefore believe our novelty is solid, and respectfully hope for your consideration.
> > >
> > > > `Q3-12` evaluation at higher scales (e.g., 8x), failure cases
> > >
> > >  `A3-12` For 8× upscaling, we compare with recent methods and report the 8x results on UDM10 below.
> > >
> > > |UDM10|PSNR↑|DISTS↓|CLIPIQA↑|DOVER↑|Ewarp↓|
> > > |-|-|-|-|-|-|
> > > |MGLD(ECCV'24)|23.97|0.1822|0.4341|0.6826|4.11|
> > > |STAR(ICCV'25)|23.72|0.2550|0.2442|0.3319|1.07|
> > > |SeedVR(CVPR'25)|23.28|0.1452|0.2942|0.6423|3.02|
> > > |SeedVR2(ICLR'26)|25.12|0.1573|0.3102|0.5373|2.33|
> > > |InfVSR(ours)|24.32|0.1416|0.4862|0.7378|2.07|
> > >
> > > The results show that our InfVSR remains **strongly competitive at 8×**, while some methods such as STAR degrade substantially under this harder setting.
> > >
> > > For **failure cases**, we provide a visualized example: the road-sign text in **Supplementary Video 1** remains difficult to fully and naturally reconstruct once it is heavily degraded. We promise to further improve the discussion of such failure cases in the Limitations section. For our claimed **stronger robustness under extreme motion**, please refer to `A2-3` for experimental support.
> > >
> > > > `Q3-13` unclear to what extent the performance gains are driven by the pretrained T2V prior
> > >
> > >  `A3-13` We believe our observed gains are not explained by the prior alone and support this with three points.
> > >
> > > 1. **Our ablations directly validate the contribution of our proposed components.** Table 4 and Supp. Sec.D shows consistent gains from AR reformulation, joint visual guidance, validated chunk/cache length, patch-wise supervision, stage-wise training and temporal constraints. **Removing any of them hurts performance.**
> > > 2. **Our method extends what the prior can do.** The backbone is pretrained on short videos, and prior T2V-based VSR works use such priors for short-form restoration, whereas our architectural and training refinements enable **streamable inference for arbitrarily long videos**, which is not provided by the prior itself.
> > > 3. **Our base model, Wan2.1-1.3B is not an unusually strong prior.** It is open-sourced, used in concurrent works such as RealisVSR (arXiv'25), OASIS (arXiv'25), and FlashVSR (CVPR'26), and relatively lightweight (Tab. 3), which is why we adopt it. The DOVE(NeurIPS'25) rebuttal (openreview.net/forum?id=DkJImu7t3A) further shows that the stronger 5B CogVideoX backbone performs better than 1.3B Wan, suggesting that our gains are not simply due to backbone strength.
> > >
> > > Therefore, we believe the observed gains are **substantially enabled by our InfVSR itself**.

---

### Official Review · Reviewer_JuG6 · 2026-03-11

**Soundness:** 3
**Presentation:** 3
**Significance:** 2
**Originality:** 2
**Overall Recommendation:** 5
**Confidence:** 3

**Summary:**

The paper proposes InfVSR, a streaming VSR method formulated as an autoregressive one-step diffusion process. The model processes videos in chunks using a causal DiT with KV-cache. During training, patch-wise supervision and cross-chunk distribution matching are introduced. Experiments show improvements in efficiency and temporal consistency metrics across several VSR benchmarks.

**Compliance With Llm Reviewing Policy:**

Affirmed.

**Final Justification:**

My concerns have been addressed in the rebuttal. This is one of the first attempts at a complete system design for the setting of streaming generative VSR. The authors acknowledge the suggested changes for the revised version based on my concerns, including evaluations across different sequence-length ranges and scene-change cases.

**Key Questions For Authors:**

1. Since the method targets long videos, how does the rolling KV-cache handle scene changes (e.g., camera cuts, large motion) where past frames may become irrelevant? Providing some visual results or comparisons for such cases would be helpful.

2. How does the method perform under varing video lengths? It would be helpful to include a length vs. performance curve under increasing sequence length with a SOTA method for reference.

3. The method also achieves best or near-best results on several datasets with short sequences. Could the authors explain which component contributes to these improvements?

**Limitations:**

As discussed in the Weakness section, my main concern is the novelty and contribution of the proposed components. Besides, it remains unclear how the proposed method handles long videos with scene changes.

**Strengths And Weaknesses:**

Strengths:

1. The proposed MovieLQ benchmark could be useful for evaluating long-form video super-resolution.
2. The idea is easy to follow. Adapting a video diffusion model to a causal DiT with a KV-cache is intuitive.
3. The experiments show consistent improvements, and the ablation studies are comprehensive and detailed.

Weaknesses:

1. Most components build on existing techniques. For example, one-step diffusion, causal DiT, KV-cache, and DMD loss [A][B] are not fundamentally new, which makes the paper appear more like an extension of existing VSR methods.
2. The chunk size is set to 3 frames for local context, and the reference frame is chosen as only the middle frame. This design is somewhat heuristic and may struggle to handle scene changes (e.g., camera cuts or large motion), where past frames could become irrelevant.
3. See questions below.

---

[A] Yin et al., “One-step Diffusion with Distribution Matching Distillation,” CVPR 2024.

[B] Huang et al., “Self Forcing: Bridging the Train-Test Gap in Autoregressive Video Diffusion,” NeurIPS 2025.

---

> ### Author Rebuttal · Authors · 2026-03-31
>
> > `Q2-1` Most components build on existing techniques. ...the paper appear more like an extension of existing VSR methods.
>
> `A2-1` Thanks for your concern. We respectfully disagree our work is merely an extension of existing VSR methods. Our work is among **the first to explore generative VSR in a practical streaming setting**, where the central challenge is no longer only frame quality, but also how to maintain fidelity, temporal consistency, and efficiency under AR settings. Our contribution is thus not to claim each component new in isolation, but to show their **non-trivial reformulation and integration** leads to a new and practical solution.
>
> More specifically, the novelty of this work lies in three aspects.
>
> First, we reformulate VSR as an **autoregressive one-step diffusion problem with streaming inference**. To the best of our knowledge, this problem setting has not been established in prior VSR methods in a **complete and practical form**. This formulation directly targets and solves the current key VSR limitation, namely their inability to scale to long videos without severe memory/runtime growth or temporal truncation.
>
> Second, the proposed framework is **not a direct combination of existing modules**. Adapting causal DiT, rolling KV-cache, and DMD to VSR is non-trivial because VSR is a restoration problem with strict fidelity constraints, rather than free-form video generation. In our method, these components are **tailored** to achieve optimal quality-efficiency balance (e.g., integration with pixel constraints, and eliminating additional forward pass for clean cache).
>
> Third, our contribution lies in the **complete system design and validation**, including the consistency-oriented architectural adaptation, training strategy, and long-video evaluation. The empirical results show that this design achieves a favorable balance between quality, consistency, and efficiency, which has not been demonstrated by existing methods.
>
> > `Q2-2` The chunk size is set to 3, and the reference frame is only the middle frame. This design is somewhat heuristic and may struggle to handle scene changes.
>
> `A2-2` First, the reference frame is not restricted to the middle frame, and it serves as complementary guidance rather than a strong constraint, so occasional weak relevance does not materially harm performance. Please refer to`A1-2` for more detailed clarification.
>
> For the chunk size, we find fixing it to 3 keeps the temporal context distribution consistent during training and inference. We test an adaptive strategy that switches the chunk length between 1 and 5 based on inter-frame change, and it leads to performance drop.
>
> |MVSR4x|PSNR|LPIPS|DOVER|Ewarp|
> |-|-|-|-|-|
> |M=3(Ours)|22.56|0.3450|0.6872|1.03|
> |1-5 adaptive|21.79|0.3567|0.6793|1.29|
>
> > `Q2-3` How does the rolling KV-cache handle scene changes? Providing some visual results or comparisons for such cases would be helpful.
>
> `A2-3` As the paper and supplementary material cannot be revised for visualizations, we manually selected five large-motion videos from VideoLQ (003,004,005,021,024) and perform comparison as follows.
>
> ||MUSIQ|CLIPIQA|DOVER|Ewarp|
> |-|-|-|-|-|
> |UAV|51.17|0.3451|0.6996|20.73|
> |SeedVR2|45.54|0.2475|0.6492|13.27|
> |Ours|54.38|0.3884|0.7172|10.14|
>
>
> > `Q2-4` length vs. performance curve.
>
> `A2-4` As suggested, we provide comparison on MovieLQ under different sequence-length ranges.
>
> |DOVER|1-200|201-400|401-600|601-800|801-1000|
> |-|-|-|-|-|-|
> |UAV|0.7963|0.8028|0.7879|0.7858|0.7913|
> |SeedVR2|0.8101|0.8056|0.7917|0.7986|0.8022|
> |Ours|0.8608|0.8594|0.8493|0.8532|0.8586|
>
>
> > `Q2-5` Which component contributes to short-sequence improvements?
>
> `A2-5` First, our method effectively leverages the pretrained **video diffusion** prior, which provides stronger generative capacity for restoring realistic details and temporal consistency beyond purely regression or T2I-based VSR models.
> Second, the **proposed dual-temporal design** is also beneficial in short sequences: the rolling KV-cache improves local smoothness, while the joint visual guidance provides stable conditioning without relying on heavy VLMs.
> Third, our **training strategy** further strengthens performance. Patch-wise supervision improves high-resolution training efficiency, and the consistency-oriented objective better balances single-frame fidelity and temporal continuity.

---

> > ### Author Rebuttal · Reviewer_JuG6 · 2026-04-03
> >
> > Thanks for your rebuttal. My concerns have been addressed, and I will consider raising the score. Overall, this is one of the first attempts at a complete system design for the practical setting: streaming generative VSR. I also encourage the authors to incorporate the suggested changes in the revised version according to the rebuttal, such as clarifying the reference frame as complementary guidance, and the analyses in the rebuttal on videos with different sequence-length ranges and scene-change cases.

---

> > > ### Author Response · Authors · 2026-04-03
> > >
> > > Thanks for your positive feedback and for recognizing our pioneering exploration of streaming generative VSR. We truly appreciate the time and effort you devoted to reviewing our paper, and we are very grateful for your constructive feedback throughout this process.
> > >
> > > Following your suggestions, we commit to revising the paper accordingly in the revised version, including:
> > >
> > > 1. clarifying the **dual role of joint visual guidance**: it serves both as a lightweight semantic prompt in place of heavy VLM-based guidance, and as shared cross-chunk prompting that improves temporal consistency;
> > > 2. clarifying our strategy for handling **large motion and scene changes**: the keyframe is not fixed, but can be updated adaptively based on the Pyscenedetect threshold;
> > > 3. supplementing the results and analysis showing that our method performs more robustly under large motion and maintains stable performance across different sequence-length ranges.
> > >
> > > Thank you once again for your recognition and for helping improve our work. We sincerely hope that the revisions will be satisfactory and support your consideration of raising the score.

---

### Official Review · Reviewer_4nKb · 2026-03-16

**Soundness:** 3
**Presentation:** 1
**Significance:** 3
**Originality:** 2
**Overall Recommendation:** 3
**Confidence:** 5

**Summary:**

The paper proposes InfVSR, an Autoregressive-One-Step-Diffusion (AR-OSD) framework designed for long-sequence Video Super-Resolution (VSR). This article mainly focus on how to maintain temporal consistency and computational efficiency when scaling generative VSR to unbounded video lengths. The authors integrate a causal DiT with rolling KV-cache and joint visual guidance, alongside a dual-level training objective combining patch-wise supervision and cross-chunk distribution matching (DMD) loss. The paper also presents a new benchmark, MovieLQ, aimed at evaluating long-form VSR under real-world degradations.

**Compliance With Llm Reviewing Policy:**

Affirmed.

**Final Justification:**

After reading the rebuttal and follow-up responses, I raised my score from 2 to 3, mainly because the authors clarified several implementation details and addressed some minor confusions. That said, I still view this submission as borderline and leaning toward rejection. The rebuttal does not sufficiently address my main concern: the limited method-level novelty and the lack of deeper mechanistic analysis beyond system integration.

While the authors argue that the work provides additional methodological insight, I do not find the current evidence and analysis convincing. I hope future revisions can provide a more principled mechanistic understanding rather than primarily intuition-level motivation and empirical observations.

In addition, regarding the first-stage rebuttal (A1-2), I remain skeptical about the reported metrics. It is highly unusual that the LR setting outperforms the SR setting, and the DAPE-based explanation is not persuasive; this issue deserves closer scrutiny.

Finally, although formatting and writing issues are not the main deciding factor, I evaluated the submission based on all provided materials (main paper + supplementary + rebuttal), and the above concerns remain.

**Key Questions For Authors:**

1. Could you clarify the specific details of the DMD implementation? Specifically, what is the exact teacher model used, and what are the detailed training configurations for Stage-2?
2. How does the model handle scene transitions or shot cuts in long videos given that the global visual guidance relies on a single, fixed Ref-image? Have you considered dynamically updating the reference image (e.g., using previously generated SR frames) to provide more relevant structural information?
3. Regarding to the loss design space, compared to Lpips loss, why do you prefer Dists-loss? Why was a simple pixel-wise temporal loss chosen instead of a motion-compensated (warp-based) loss? In scenarios with large motion or scene transitions, how does the model prevent $L_{temp}$ from introducing blur? Could you provide evidence that the hyperparameter for this loss is robust across different motion intensities. Furthermore, Table 4(b) indicates that the numerical gains from this module are marginal.
4. The proposed auto-regressive method is more likely to chunk-wise AR. As reported in Table 4(c), simply increase M could be helpful, which is somewhat obvious. My main concern is could some training techniques solve this, e.g. your temp-loss or DMD. Otherwise, it may still sound incremental.

(Note: I am open to reconsidering my evaluation and raising the final score if the authors can provide the missing implementation details for DMD and adequately address the concerns regarding the evaluation metrics and the mechanisms of the proposed modules during the rebuttal phase.

**Limitations:**

Yes.

**Strengths And Weaknesses:**

Strengths.
1. This paper discusses an under-explored bottleneck in VSR application (scale to long video and huge computation cost). Using distillation method and efficient training strategies to address this problem.
2. While the individual components are not entirely novel, the paper demonstrates a solid engineering effort by effectively combining popular and state-of-the-art techniques(e.g. an Autoregressive inference pipeline, rolling KV-cache, and DMD), into a cohesive and highly efficient framework for streaming VSR.

Weakness.
1. The evaluation of low-level temporal consistency is insufficient. For VSR tasks, metrics like VBench focus more on semantic-level consistency (which LR video already persists), while the optical flow-based warping error ($E_{warp}^{*}$) is inherently limited by the estimation errors of the flow network itself. The field requires metrics that directly assess texture-level temporal continuity. Consequently, the proposed MovieLQ benchmark is primarily valuable for testing duration limits rather than providing a strict, robust assessment of temporal consistency.
2. The design of using a fixed global reference image (Ref-image) is flawed for long-form videos. In real-world long videos, scene changes and shot cuts are frequent; a fixed reference frame becomes invalid once the scene transitions. Furthermore, the effectiveness of this reference image is questionable. The ablation study in Table 4(b) shows marginal numerical improvements across metrics. It is also unexplored whether using the generated SR image as a dynamic reference would provide more informative guidance than the LR image, since artifacts mainly come from diffusion process in different video frames.
3. The implementation details regarding the cross-chunk distribution matching are unclear. Specifically, the identity and configuration of the teacher model are not explicitly defined in the text. The specific operational details of the Stage-2 training phase also lack sufficient description, making it difficult for readers to reproduce the AR training process, e.g. how the teacher model init?
4. The paper presents a highly effective system-level engineering integration. However, it currently lacks the necessary mechanistic or theoretical discussion regarding the proposed framework.(e.g why DMD loss decrease $E_{warp}^{*}$, asymmetric distillation should improve consistency? \cite{1} ). Enhancing the methodological or analytic depth would be more suitable for ICML.

[1] Yin, Tianwei, et al. "From slow bidirectional to fast autoregressive video diffusion models." Proceedings of the IEEE/CVF Conference on Computer Vision and Pattern Recognition. 2025.

---

> ### Author Rebuttal · Authors · 2026-03-31
>
> Thank you for your valuable comments and openness to reconsideration. We are concerned that some implementation details are provided in the supplementary material and may be overlooked. We hope our responses below help clarify these points and support your reconsideration.
>
> > `Q1-1` The evaluation of low-level temporal consistency is insufficient.
>
> `A1-1` Inherently, temporal consistency in VSR is difficult to characterize with a single perfect metric. This is a broader issue in I/VSR: PSNR/SSIM favor fidelity but often prefer over-smoothed results, while perceptual metrics better reflect visual quality yet still do not fully align with human judgment. Therefore, restoration quality is typically assessed through **complementary metrics rather than any single metric alone**.
>
> Our evaluation follows this principle: warping-based metrics are more sensitive to low-level alignment, but can favor smoother textures that are easier to align. Semantic consistency metrics are therefore also necessary to verify that temporal stability is achieved together with preserved identity and coherent appearance.
>
> We also clarify that **semantic consistency is not fully preserved by the LR input**. VBench scores of LR and ours below prove our model actively improves consistency, and evaluating semantic consistency is valuable.
>
> |(SC+BC+MS)/3|UDM10|SPMCS|VideoLQ|
> |-|-|-|-|
> |LQ|0.9405|0.9647|0.9391|
> |Ours|0.9688|0.9811|0.9562|
>
> Beyond metrics, we also provide abundant video results in the supplementary material as the most direct and compelling evidence.
>
> > `Q1-2` The design of using a fixed global reference image is flawed for long-form videos.
>
> `A1-2` As clarified in Supplementary Sec. A.3, **the reference image is not strictly fixed**: we apply threshold-based switching. Different thresholds trade off finer prompt updates against fewer prompt extractions, but all **improve performance** over not using prompts (Tab.4b a vs.c). This also reflects the design logic of this module which we need to rejustify: **its primary purpose is to provide a lightweight (0.2s) alternative to heavy prompting modules such as VLMs**, while its secondary benefit is that sharing prompts across neighboring chunks leads to better consistency. Therefore, this module serves as lightweight enhancement rather than strong constraint, which we believe is both meaningful and practical. We will clarify this mechanism further in the revision.
>
> As suggested, we further compare SR/LR reference frames, and find similar results, which supports our original design choice of using degradation-aware DAPE.
>
> |UDM10|PSNR|LPIPS|DOVER|Ewarp|
> |-|-|-|-|-|
> |w/SR|24.79|0.2970|0.7856|1.96|
> |w/LR(Ours)|24.86|0.2972|0.7826|1.95|
>
> > `Q1-3` The implementation details regarding the cross-chunk DMD are unclear.
>
> `A1-3` Supplementary Sec. B provides the complete Stage-2 training pseudocode; please refer to it. The teacher is initialized from the original Wan2.1-1.3B, following common SR practice.
>
> > `Q1-4` ...Enhancing the methodological or analytic depth would be more suitable for ICML.
>
> `A1-4` Thanks for the insightful suggestion. Our paper is not intended as purely system-level engineering, but is built on a set of **methodological insights** into streaming generative VSR: temporal consistency requires both short-term and long-term modeling, and should be enforced from both pixel-level and semantic-level perspectives.
>
> For the specific point raised, DMD encourages the AR student to learn the 3D spatiotemporal distribution of pretrained video diffusion, thus improving semantic consistency, as reflected by SC/BC. Meanwhile, the richer textures it produces are harder to align than over-smoothed results, which can make Ewarp higher. This further shows why **semantic metrics** are needed as a complementary view.
>
> > `Q1-5` Why  Dists-loss? Why $L_{temp}$? How to prevent $L_{temp}$ from introducing blur? Prove this loss is robust across motion intensities.
>
>  `A1-5`  DISTS is used as a standard perceptual loss following DOVE, without special treatment. We find $L_{temp}$ is more effective; see Supplementary Sec. D for ablations. Under large motion, temporal differences increase in both GT and prediction, so **the constraint remains valid** and does not inherently introduce blur. By contrast, flow-based losses are more sensitive to inaccurate optical flow. We further supplement an ablation on VideoLQ, which contains stronger motion, and this loss still improves consistency.
>
> |VideoLQ|DOVER|Ewarp|
> |-|-|-|
> |w/o $L_{temp}$|0.7514|9.94|
> |w/ $L_{temp}$(Ours)|0.7556|7.52|
>
> > `Q1-6` Simply increase M could be helpful ... Could some training techniques solve this?
>
>  `A1-6` Yes, and this is exactly our key point: under the small M required for streaming deployment, the goal is to maintain consistency through training design. Accordingly, we introduce a novel dual modeling and dual regularization design to address this challenge, thereby achieving the best efficiency–performance trade-off.

---

> > ### Author Rebuttal · Reviewer_4nKb · 2026-03-31
> >
> > Thank you for the detailed rebuttal and the additional experimental results. The response addressed several of my questions and clarified the intended scope of the method. I will adjust my score accordingly.
> >
> > However, I still do not believe this paper should be accepted to ICML in its current form. My main concern remains that the work lacks sufficient method-level analysis and insight. In particular, the newly added SR results in the rebuttal appear somewhat counter-intuitive and potentially inconsistent with observations from concurrent works (e.g., SparkVSR). I would have expected a clearer principled explanation of why the proposed design leads to such behavior, rather than additional numerical comparisons alone.
> >
> > Moreover, I do not think the newly introduced benchmark should be framed as a core contribution. As the authors also acknowledged (A1-1), collecting longer videos by itself does not constitute a strong research contribution unless accompanied by a clearly motivated evaluation protocol and analysis demonstrating what fundamentally new failure modes it reveals.
> >
> > Finally, the submission formatting should follow ICML conventions: the appendix is expected to appear after the main paper rather than being placed solely in supplementary material, and key claims should be explicitly referenced from the main text.
> >
> > Overall, the paper is reasonably well-executed and the combination of techniques is competitive compared to recent related works (e.g., DUOVSR, FlashVSR). However, it remains difficult to extract clear conceptual insights from the proposed approach, and I therefore maintain a weak reject recommendation.
> >
> > [1] Yu, J., Gao, X., Verlani, P., Gadde, A., Wang, Y., Adsumilli, B., & Tu, Z. (2026). SparkVSR: Interactive Video Super-Resolution via Sparse Keyframe Propagation. arXiv preprint arXiv:2603.16864.
> > [2] Lv, Z., Xia, M., Wang, X., & Wong, K. Y. K. (2026). DUO-VSR: Dual-Stream Distillation for One-Step Video Super-Resolution. arXiv preprint arXiv:2603.22271.
> > [3] Zhuang, J., Guo, S., Cai, X., Li, X., Liu, Y., Yuan, C., & Xue, T. (2025). Flashvsr: Towards real-time diffusion-based streaming video super-resolution. arXiv preprint arXiv:2510.12747.

---

> > > ### Author Response · Authors · 2026-04-03
> > >
> > > **[Update for Final Justification]**
> > >
> > > Thanks for your feedback in final justification.
> > >
> > > > Regarding results in `A1-2`:
> > >
> > > (1) We honestly report results;
> > >
> > > (2) LR doesn't strictly outperform SR, but is only very close;
> > >
> > > (3) The DOVER increase and PSNR decrease reflect SR's slight increase in high-frequency details, which is intuitively plausible.
> > >
> > > (4) Using SR as a prompt in our setting yields marginal gains that don't justify the extra computational cost. So we choose LR as guidance.
> > >
> > > (5) Such design tradeoff still carries methodological value beyond mere engineering integration, as also recognized by **Reviewer JuG6** ('This is one of the first attempts'), **Reviewer hLcA** ('improved the experimental validation to address my concerns'), and **Reviewer WGFo** ('my concerns have been addressed').
> > >
> > > > Regarding novelty, we clarify it again.
> > >
> > > We have clarified the novelty with extensive results (see `A1-2`, `A2-2`, `A4-5`). It provides principled mechanistic understanding of the novelty of our method, which is also acknowledged in Final Justification and Rebuttal Acknowledgement by **Reviewer JuG6**, **Reviewer hLcA** (also see `A3-11`), and **Reviewer WGFo**.
> > >
> > > ---
> > >
> > > > `Q1-7` should not be accepted to ICML ... the work lacks sufficient method-level analysis and insight
> > >
> > > `A1-7` First, we believe ICML is not limited to “conceptual novelty”. Its CallForPapers explicitly includes **theory, evaluation, systems, and application-driven ML**. And  ReviewerInstructions notes that originality may arise from **creative combinations of existing ideas, real-world use cases, or removing restrictive assumptions**.
> > >
> > > We also believe we have method-level analysis and insight, and clarify our contribution as follows:
> > >
> > > ||Practical bottleneck|Insight|Solution Method|Why this is non-trivial|
> > > |-|-|-|-|-|
> > > |1|Long-video generative VSR is hard to scale due to memory/runtime growth|VSR requires a causal reformulation|AR-OSD paradigm|First complete and formal reformulation|
> > > |2|Temporal consistency cannot be handled by a single mechanism|Consistency requires short-range continuity and long-range semantic anchoring|rolling KV-cache + lightweight global semantic guidance + patch pixel supervision + DMD asymmetric distillation|First consistency-driven streaming framework tailored for efficient generative VSR|
> > > |3|Pixel-level evaluation alone tends to favor smoother solutions|Evaluate from both pixel and semantic level perspectives|Complementary metrics|Reveal single-metric missing findings|
> > > |4|Short benchmarks under-reveal deployment-time temporal failures|Long-horizon evaluation is needed|MovieLQ|Make long-horizon deployment measurable|
> > >
> > > > `Q1-8` a clearer principled explanation of why the proposed design leads to such behavior
> > >
> > > `A1-8` The key point is that our design is effective but **different in paradigm** from SparkVSR-style reference conditioning.
> > >
> > > |Aspect|SparkVSR-style reference conditioning|Ours|
> > > |-|-|-|
> > > |Guidance form|VAE-encoded keyframe latent|DAPE semantic features|
> > > |Injection manner|in-context conditioning|injected into the original text cross-attention|
> > > |Constraint strength|strong, detail-level constraint|soft semantic anchoring|
> > > |Computation overhead|heavy, with addtitional encoding/interaction/token cost|lightweight and OSD-compatible|
> > > |Performance gain source|introducing high-quality reference priors|introducing multimodal (semantic) guidance|
> > >
> > > This difference also explains **why using SR/LR leads to similar results in our setting**. First, our module provides lightweight semantic prompting rather than strong detail-level conditioning, so SR/LR frames offer similar semantic cues and lead to naturally close effects. Second, DAPE is finetuned to align LR and SR representations, which further reduces the gap between using LR and SR as reference.
> > >
> > > We will revise the main paper to further emphasize that this module is not designed to strongly constrain the output, but rather to (1) provide lightweight semantic guidance in place of heavy VLMs, and (2) share prompts across neighboring chunks rather than using independent prompts, so as to reduce extraction cost and improve consistency.
> > >
> > > > `Q1-9` do not think the newly introduced benchmark should be framed as a core contribution ... what fundamentally new failure modes it reveals
> > >
> > > `A1-9` **Clearly motivated evaluation protocol**: the complementary use of semantic-level and pixel-level metrics reveals behaviors that cannot be captured by any single metric alone.
> > >
> > > **What fundamentally new failure modes it reveals**: in practical deployment, other methods often rely on **temporal breakdown + fusion**, which makes it difficult to maintain continuity over long horizons (see Fig. 1). Our InfVSR processes the sequence in one streaming pass. This benchmark makes that difference directly measurable.
> > >
> > > > `Q1-10` submission formatting should follow ICML conventions
> > >
> > > `A1-10` We will correct this in revision. We kindly hope the supplementary file may still be considered for clarification.

---

### Decision · Program_Chairs · 2026-04-30

**Decision:**

Accept (regular)

**Comment:**

This paper was reviewed by 4 experts in the field. After discussion, the reviewers still hold a mixed review to this work. The rating is 3 (Weak Reject), 4 (Weak Accept), 5 (Accept), 5 (Accept).

On the positive sides, reviewers agree that this work 1) provides a well-motivated AR-OSD formulation for streaming consistency, and 2) achieves high computational efficiency and significant speed-ups.

Still, reviewers raised several concerns to this work. The concern includes: 1) limited conceptual novelty as the framework is an integration of existing components, 2) insufficient low-level temporal consistency evaluation and very limited qualitative evaluation. While area chair think this work does introduce non-trivial design to this problem, the evaluation is problematic. Even with significant improvement over existing methods in the reported number, it is actual performance of the proposed solution is still unclear (lack of convincing temporal stability evaluation, no user study and no VBench result on SeedVR2).

Based on this, the decision of this work is to "weak acceptance (low priority: accept if there is room in the program)". The proposed one-step streaming solution is seemingly effective and does demonstrate in-depth to the problem. However, the main reason not to give higher score is the uncertainty of evaluation.

We strongly recommend the authors carefully read all reviewers’ final feedback and revise the manuscript as suggested in the final camera-ready version if being accepted.